# Urban underlying surface modulates summertime thunderstorm processes and associated lightning activities

Tao Shi[1,2], Yuanjian Yang[3], Gaopeng Lu[4], Zuofang Zheng[5], Yucheng Zi[4], Ye Tian[6], Lei Liu[3], and Simone Lolli[7]

[1]School of Mathematics and Computer Science, Tongling University, Tongling, 244000, China
[2]Key Laboratory of Transportation Meteorology of China Meteorological Administration,
Nanjing Joint Institute for Atmospheric Sciences, Nanjing, 210041, China
[3]State TS1 Key Laboratory of Climate System Prediction and Risk Management,
Nanjing University of Information Science and Technology, Nanjing, 210044, China
[4]School of Earth and Space Sciences, University of Science and Technology of China, Hefei, 241000, China
[5]Institute of Urban Meteorology, China Meteorological Administration, Beijing, 100000, China
[6]Beijing Meteorological Observation Center, Beijing, 100000, China
[7]CNR-IMAA, Contrada S. Loja, 85050 Tito Scalo (PZ), Italy

**Correspondence:** Yuanjian Yang (yyj1985@nuist.edu.cn)

**Abstract.** The urban underlying surface may have a significant impact on thunderstorm processes and lightning activities, but there is still a lack of explanation of the mechanism. Through a comparative analysis of cloud-to-ground (CG) lightning location datasets in three cities of varying sizes in China, it was observed that for small cities, CG activities tend to cluster towards the city center, whereas in large cities, CG activities tend to accumulate in the city periphery. Radar echoes indicated the occurrence of a significant barrier effect as the thunderstorm that occurred on 13 July 2017 ("0713" case) passed over Beijing's built-up area. An analysis of ground observations revealed that when this thunderstorm passed over the rough urban underlying surface, a separation of the near-surface cold pool emerged. This separation led to the weakening of vertical airflow and the breakdown of the convergence line, ultimately triggering the bifurcation and movement of the thunderstorm. The Weather Research and Forecasting (WRF) numerical simulations have facilitated our further exploration into the potential mechanism of the barrier effect. When a portion of the built-up area was replaced with bare land, the separation of the cold pool and the breakdown of the convergence line were notably mitigated. Additionally, the building density could also influence the evolution of the cold pool and convergence line. Consequently, the urban underlying surface might be a potentially crucial factor affecting the thunderstorm processes and CG activities. Our findings provide crucial scientific insights for refined forecasting, early warning, and risk assessment of lightning disasters, strategy formulation for urban disaster prevention and mitigation, and resilient city planning and development.

## 1 Introduction

As urbanization progressed, lightning events emerged as a significant hazard to city safety and social development, presenting a serious weather-related risk (Westcott, 1995; Pinto et al., 1999). Among these events, CG lightning poses the greatest risk to both ground-based objects and populations. The scientific community has widely acknowledged the consensus regarding the enhancement of the urban thermal effect on thunderstorm processes (Shepherd, 2005; Wang et al., 2018, 2021; Yue et al., 2019; Shi et al., 2023). The thermal effect increases the boundary layer height above cities and the vertical mixing height (Xu et al., 2013; Sun et al., 2021; Shi et al., 2025). The vertical mesoscale cutting resulting from temperature differences between urban and suburban areas serves as a key prerequisite for the development of the convective system (Farias et al., 2014; Sun et al., 2013).

Previous studies have also recognized the dynamic effect of the urban underlying surface on thunderstorm processes and lightning activities (Bornstein and LeRoy, 1990; Dai et al., 2005; Yang et al., 2021; Shi et al., 2023). Scholars have explored the dynamic effects of urban underlying surfaces through numerical simulations. The results from global and regional climate models show that the urban rough underlying surface can alter the horizontal wind field, enhancing convergence and upward movement in the upstream direction (Jin and Shepherd, 2005), which, to some extent, facilitates the development of thunderstorm systems (Yin et al., 2020). The "climbing" upward airflow movement, as simulated by the WRF model (Zhu et al., 2016), exhibits a relatively weak intensity, insufficient to alter the organizational processes of thunderstorms significantly. Moreover, through urban boundary layer model simulations, researchers have also discovered that when thunderstorms pass over cities, the dynamic effect of the urban underlying surface can lead to the bifurcation and movement of thunderstorm systems (Bornstein and LeRoy, 1990). This phenomenon is known as the barrier effect (Stallins and Bentley, 2006). The aforementioned simulation work has made valuable explorations into studying the dynamic effects of urban underlying surfaces. However, the current research only employs simulation scenarios with and without an urban underlying surface, without delving into the detailed characteristics of urban spatial configuration. It is worth noting that, within the spatial configuration of the urban underlying surface, city size is recognized as a crucial factor impacting thunderstorm processes (Kingfield et al., 2018). Additionally, building density also demonstrates a tendency to alter urban lightning activities (Stallins and Bentley, 2006). Therefore, it is necessary to comprehensively consider the spatial configuration characteristics of the urban underlying surface and continue to explore the influential mechanisms of the urban underlying surface on thunderstorm processes and associated lightning activities.

With a population exceeding 20 million, a built-up area spanning approximately 1500 km², and a gross domestic product (GDP) exceeding 4 trillion, the Beijing megacity stands as the most urbanized city within the urban agglomeration of Beijing–Tianjin–Hebei (National Bureau of Statistics of the People's Republic of China, 2021). In recent years, the Beijing megacity has been repeatedly impacted by severe convective events, leading to significant economic and social disruptions (Qie et al., 2021). Some studies analyzing thunderstorms passing over the Beijing megacity have found that the urban barrier effect primarily dominates thunderstorm processes (Dou et al., 2015; Shi et al., 2023). However, other researchers pointed out that, despite the buildings in the Beijing megacity splitting the squall line into convective cells, the barrier effect on convection is less significant compared to the thermal effect (Miao et al., 2011). In general, there is an ongoing debate regarding the dominant role of Beijing's urban underlying surface on summertime thunderstorm processes, with potential thermodynamic mechanisms remaining a gap in research.

To address these issues, this study focused on the Beijing megacity as the main research area, utilizing radar echo data and lightning location data to analyze the organizational processes and the spatial patterns of CG activities during thunderstorms passing over the city. Subsequently, numerical simulations were employed to explore the mechanisms by which the urban underlying surface influenced the evolution of the thermodynamic structure of thunderstorms. Our present work aims to contribute valuable theoretical insights and technical support to enhance the prediction, nowcasting, warning, and risk assessment capabilities for urban thunderstorm disasters.

## 2 Data and methodology

### 2.1 Study area

The Beijing megacity serves as the political, economic, cultural, and scientific center of China. With a dense population and rapid urbanization, the built-up area of Beijing has expanded to more than 1500 km², covering most of the southeastern plain regions (as shown in Fig. 1). Thunderstorms in Beijing typically originate from the western mountains and spread to the northeast and southeast plains (Chen et al., 2012). When interacting with warm and humid airflows southward, these thunderstorms often intensify and form squall lines (Sun and Yang, 2008; Xiao et al., 2017). Zhangjiakou and Tianjin are located northwest and southeast of Beijing. Both regions show a similar climate to Beijing, characterized by a temperate continental monsoon climate with frequent thunderstorms during the summer. Zhangjiakou is located approximately 180 km from Beijing, with a built-up area of 104.2 km², which is only 1/10 the size of Beijing's built-up area. It is considered to be one of the least urbanized cities within the Beijing–Tianjin–Hebei urban agglomeration. In comparison, the built-up area of Tianjin, situated just 60 km from Beijing, spans approximately

$605.42\,\text{km}^2$. Beijing, Zhangjiakou, and Tianjin show different urbanization levels. For this reason, they were used to explore the effects of the urban underlying surface on lightning and storm processes.

## 2.2 Data

The State Grid Lightning Network (SGLNET) is utilized to collect lightning location datasets, which include longitude and latitude, GPS time, peak current, polarity, and other relevant information (Wang et al., 2021). Lightning events are detected using magnetic direction finding and time-of-arrival (MDF-TOA) technologies within the SGLNET, achieving a detection efficiency of 94 % and a location error of 489 m (Chen et al., 2012). A previous study by Orville et al. (2002) pointed out that intra-cloud (IC) discharges may contaminate the CG lightning detection network. To address this issue, we used a screening criterion based on a peak current threshold of less than 10 kA to eliminate the potential interference from IC discharges (Schulz et al., 2005). In this study, the SGLNET lightning location data from the summer months (June–August) between 2010 and 2017 were selected to analyze the characteristics of urban CG lightning activities.

The hourly observation data from the auto weather station (AWS) utilized in this study were obtained from the China Meteorological Data Service Center (http://data.cma.cn/en, last access: 1 February 2025). This dataset included near-surface air temperature, wind speed, and wind direction. To address missing values in the observation sequence, we used a method previously described by Yang et al. (2011). Specifically, we replaced missing values with the average of synchronous observation data from the nearest five stations, and stations with excessive erroneous records were excluded. For this study, we selected 54 AWSs, which are evenly distributed throughout the built-up area of Beijing, to analyze the temporal-spatial pattern of the near-surface thermal-dynamic field.

This radar observation system consists of a data acquisition subsystem, a product generation subsystem, and a main user terminal subsystem. It enables real-time data transmission and image stitching, significantly boosting the monitoring and early warning capabilities for disastrous weather conditions such as severe convective weather, tropical cyclones, and heavy rainfall. The radar data employed in this paper are the composite reflectivity (CR) product generated by the S-band Doppler radar stationed at the Beijing Nanjiao Observatory. Previous studies have consistently recognized a threshold of 35 dBZ as a pivotal marker signifying the presence of a convective echo (Dixon and Wiener, 1993; Roberts and Rutledge, 2003; Mecikalski and Bedka, 2006). Consequently, this research adopts this well-established reflectivity threshold as the criterion for identifying thunderstorms. In addition, to gain a broader understanding of the synoptic background of thunderstorms, we utilized sounding data from the Beijing Nanjiao Observatory. These sounding data were collected at 02:00, 08:00, 14:00, and 20:00 Beijing time (BJT) every day.

Stewart and Oke (2012) introduced the concept of local climate zone (LCZ) datasets, which refer to areas with identical land use, similar spatial morphology, building materials, and human activities, on a scale ranging from a few hundred to a few thousand meters. The LCZ datasets used in this paper were provided by the Institute of Urban Meteorology, China Meteorological Administration (as shown in Fig. S1 in the Supplement). Detailed descriptions and attribute parameters of categories of LCZ datasets were presented in Table S1 in the Supplement.

## 2.3 Numerical simulation scheme

Currently, the WRF model holds a prominent position as the primary tool for simulating urban environments. The WRF model, collaboratively developed by institutions such as the National Center for Atmospheric Research (NCAR) and the National Centers for Environmental Prediction (NCEP) in the United States, is primarily designed for operational forecasting and atmospheric research. This model enables researchers to simulate real-world or hypothetical scenarios computationally, offering a highly flexible and efficient predictive framework that can be applied to studies examining the impacts related to urban meteorology (Chen and Wang, 2012). The WRF model, when coupled with the Urban Canopy Model (UCM), is employed to describe the dynamic, thermal, and radiative interactions between urban land surface processes and the upper atmosphere (Kusaka et al., 2001; Chen and Wang, 2012). The UCM not only accounts for the geometric measurements of urban buildings and roads but also optimizes the physical parameters of urban canopies. Furthermore, it calculates heat transfer across building roofs, walls, and road surfaces. The UCM is widely utilized in studies examining atmospheric boundary layer processes and environmental issues related to urbanization (He et al., 2019). Although the WRF-UCM cannot directly simulate lightning, it accurately resolves urban surface configurations, modifying boundary layer conditions and thermodynamic structures during thunderstorms, thereby enabling the analysis of urban surface impacts on lightning activity.

In this study, the WRF 4.0 version, integrated with UCM, was configured to use a triple-nested grid system with horizontal resolutions of 5 km, 1 km, and 200 m, respectively (Fig. 2). The model center is located at (40.0° N, 116.6° E), with grid dimensions of $515 \times 151$, $256 \times 251$, and $506 \times 501$, along with 38 vertical layers. The underlying surface data encompassed land use and urban canopy datasets with a resolution of 10 m. These data were used to accurately describe urban morphological features and were matched to the nested grid resolution before being implemented in the UCM. This model employed the WRF Single Moment 6-class microphysical process scheme (WSM6), the rapid radiative transfer longwave radiation scheme (RRTM), the Dudhia short-

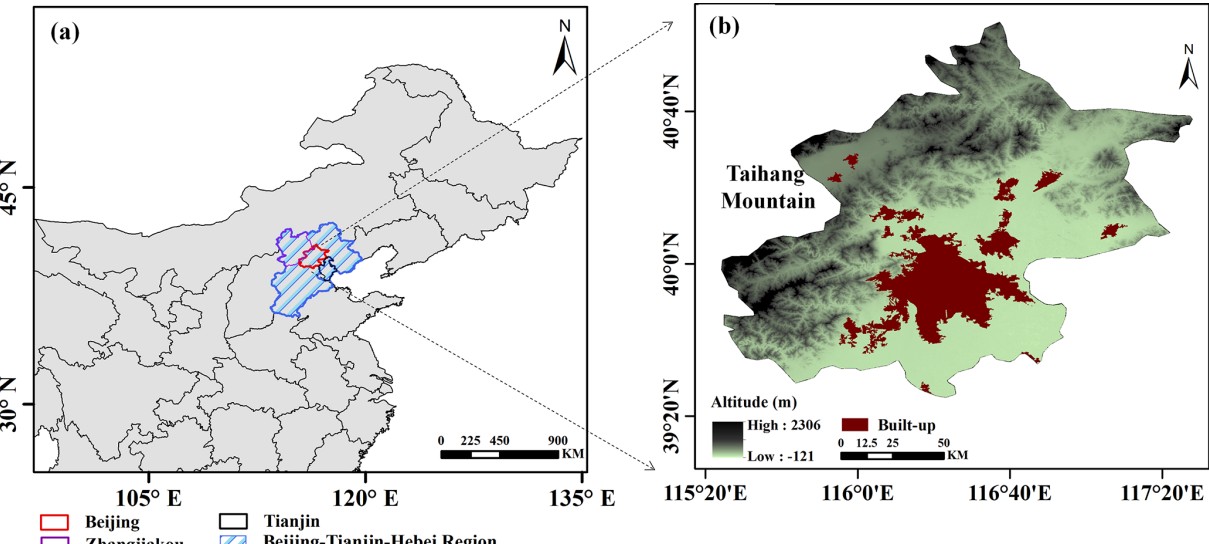

**Figure 1.** Overview of the study area. **(a)** Geographical location of the study area (Beijing–Tianjin–Hebei region) within China, identifying the cities of Beijing, Zhangjiakou, and Tianjin. **(b)** Topographic relief (including the Taihang mountains) and spatial distribution of built-up areas in the Beijing megacity, with the altitude variation depicted by the color gradient.

wave radiation scheme, the step-mountain similarity theory near-surface layer scheme, and the BouLac boundary layer scheme (Lim and Hong, 2010; Lacono et al., 2008; Janjic, 1994; Melin et al., 2017; Tewari et al., 2004).

Among all thunderstorms passing through the built-up area, the 13 July 2017 event (Case "0713") showed the strongest cloud-to-ground (CG) lightning activity, with a peak rate of 811.6 flashes h$^{-1}$. Notably, most CG flashes occurred at the urban periphery. This paper therefore selected Case "0713" for numerical simulation, running from 00:00 UTC on 13 July for 24 h. The length and area of the 5th Ring Road (RR) in Beijing are approximately 98.6 km and 600 km$^2$, respectively. To more accurately assess the impact of the urban underlying surface on the dynamic and thermal effects, we conducted five sets of experiments. These included a control experiment (EXP1) and four sensitivity tests (EXP2, EXP3, EXP4, EXP5). The specifics of these experiments are described in Table 1.

## 3   Results

### 3.1   CG activities around the built-up area

Spatial analysis of long-term lightning location data offers valuable insights into the climatic patterns of CG activities, which constitute a crucial component of urban thunderstorm disaster research. In this section, a thorough examination of the spatial characteristics of CG activities was conducted in the built-up area of Beijing using SGLNET data. Additionally, this section introduces Zhangjiakou and Tianjin, which have similar climatic backgrounds, with the aim of exploring the potential impact of different city sizes on lightning activity.

Regarding the color bar, we chose different limits for each city due to the significant variation in CG density among them. This approach allows for a clearer observation of internal lightning distribution characteristics within each city. The locations and built-up area scales of Beijing, Zhangjiakou, and Tianjin are illustrated in Fig. 3a. SGLNET recorded more than 50 000 CG flashes in the built-up area of Beijing during the study period. It is clear that the distribution of CG density in the built-up area of Beijing was uneven (Fig. 3b). There were clusters of abundant CG flashes visible and concentrated upwind and downwind of the built-up area (marked as H), with an average and maximum CG density reaching 4 and 6 fl km$^{-2}$, respectively. It should be noted that the CG flashes in the city center were sparse (marked as L), with an average CG density of less than 1 fl km$^{-2}$. This pattern was similar to the spatial distribution of CG flashes observed in the United States, more precisely in Houston (Steiger et al., 2002) and Atlanta (Stallins and Bentley, 2006). Therefore, we could speculate that there existed a potential barrier effect that alters the pattern of CG lightning in the built-up area of Beijing. As shown in Fig. 3c, the entire built-up area of Zhangjiakou (ZJK) was mainly covered by a high CG density (H) region, with a peak density of 3.6 fl km$^{-2}$ in the city center. Examining Fig. 3d, it was evident that a significant concentration of CG lightning occurs along the outskirts of the built-up area of Tianjin (TJ), with a maximum CG density of 5.2 fl km$^{-2}$. Similar to Beijing, the center of the built-up area in Tianjin exhibited a low CG density (L) region. Thus, this paper could infer that under similar climatic conditions,

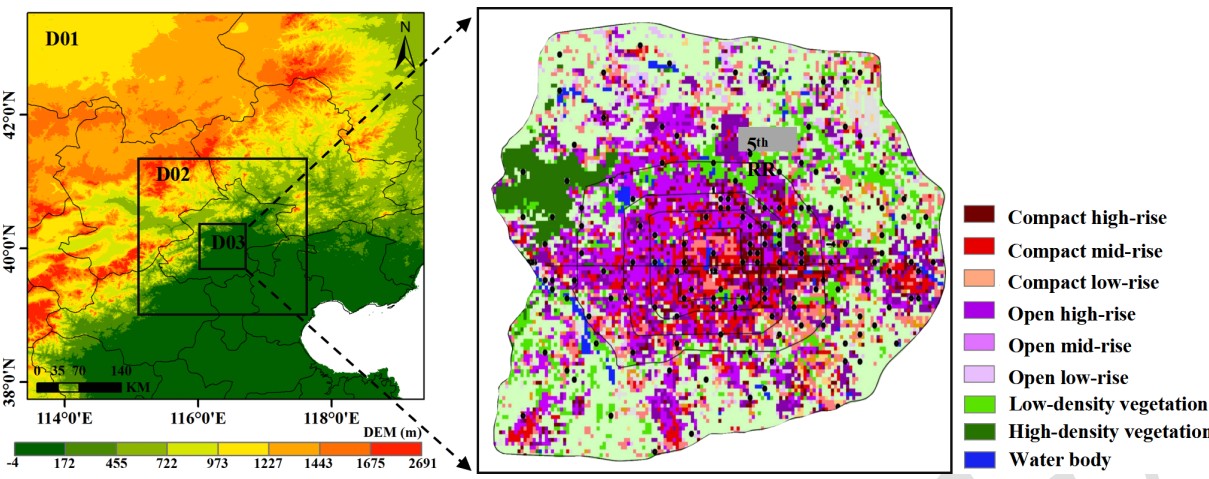

**Figure 2. (a)** Topographic distribution across the WRF triple-nested domains (D01, D02, and D03). **(b)** Spatial classification of the urban configuration structure, illustrating compact/open rise differentiated by height levels (high, mid, and low) alongside vegetation and water body categories.

**Table 1.** Description of the different experiments.

| Experiment | Style | Description |
| --- | --- | --- |
| EXP1 | Controlled experiment | Actual land use situation and building structure |
| EXP2 | Sensitive experiment | Outside the 5th RR with only bare land |
| EXP3 | Sensitive experiment | The built-up area with only bare land |
| EXP4 | Sensitive experiment | The built-up area with only open rise |
| EXP5 | Sensitive experiment | The built-up area with only compact rise |

the larger the built-up area, the lower the CG density in the city center.

The impact of building density on CG activities could also not be ignored. Stallins and Bentley (2006) used GIS technology to analyze the distribution characteristics of lightning in Atlanta, USA, and discovered that the lightning density was low in high-density building areas. This section summarized the spatial distribution of CG density and various LCZs in Beijing, aiming to understand the relationship between CG activities and the urban underlying surface. The statistical results of CG density for different types of LCZs were illustrated in Fig. S2 in the Supplement. The highest average CG density is observed in LCZ1, with a value of $3.7\,\mathrm{fl\,km^{-2}}$, while the lowest average CG density is found in LCZ6, at $2.8\,\mathrm{fl\,km^{-2}}$. In Fig. 4a, density information was used as the dominant factor to classify the urban configuration structure, which was divided into compact rise (LCZ1, LCZ2, and LCZ3) and open rise (LCZ4, LCZ5, and LCZ6). The edges of the built-up area in Beijing are primarily composed of open rise. The city center is completely constituted of compact rise, forming large-scale compact rise clusters with an area of approximately $100\,\mathrm{km^2}$ (depicted in the black box), which was largely congruent with the low CG density areas within the built-up area. Figure 4b TS2 illustrates the distribution of various LCZs in the high CG density areas (H)

and low CG density areas (L). The H areas were primarily located in the upper and lower wind directions, spanning over $1000\,\mathrm{km^2}$. Among these areas, LCZ5 was the largest building type, comprising 14 524 pixels and representing 31.2 % of the total H areas. The areas of other LCZs were relatively similar, accounting for 11.05 % − 21.2 %. The L areas were primarily concentrated in the city center, spanning a total area of approximately $90\,\mathrm{km^2}$. This area comprised 9936 dense pixels, representing 91.8 % of the total L areas. The L areas were mainly composed of compact rise and contained very few open rise. Consequently, the city size and building density might be important factors affecting CG activities.

## 3.2 The evolution of the thunderstorm passing over the urban underlying surface

The CR product detected by the Beijing Nanjiao Observatory was utilized to analyze the evolution of the "0713" case passing over the built-up area.

Figure 5a depicted the "0713" case, a quasi-linear convective system consisting of multiple cells that propagated from Hebei at 20:00 BJT. At this time, the thunderstorm system reached the northwest of the built-up area. Thunderstorm cells were continuously generated, developed, and merged on the right rear side of the thunderstorm system. At 21:00 BJT

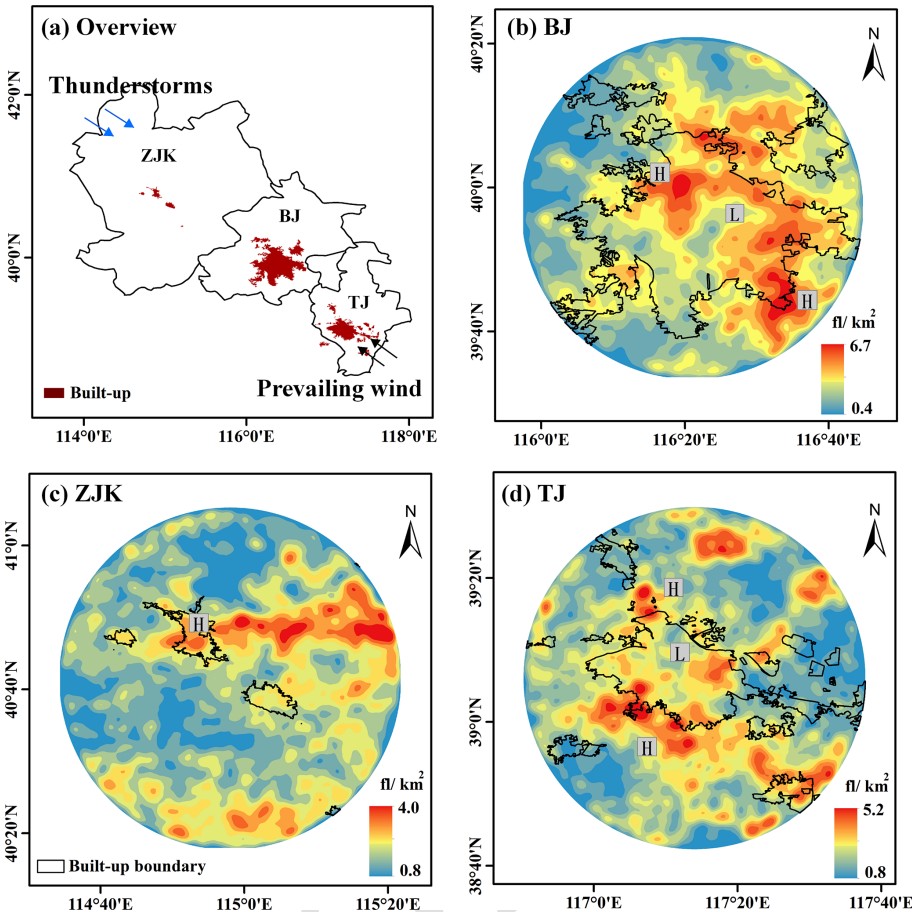

**Figure 3. (a)** Built-up area distribution across Beijing (BJ), Zhangjiakou (ZJK), and Tianjin (TJ), with the thunderstorm occurrence and prevailing wind direction indicated. **(b–d)** Spatial patterns of CG density within the built-up areas of Beijing **(b)**, Zhangjiakou **(c)**, and Tianjin **(d)** during the summer of 2010–2017.

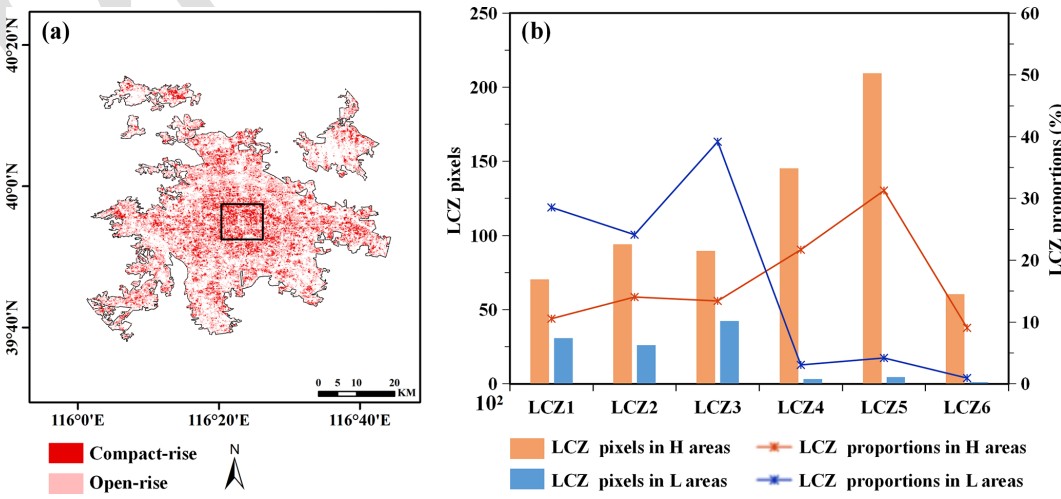

**Figure 4. (a)** Characteristics of the urban configuration structure dominated by density information, including compact rise (LCZ1, LCZ2, and LCZ3) and open rise (LCZ4, LCZ5, and LCZ6). **(b)** The configuration structure of buildings in areas with high CG density (H) and low CG density (L).

(Fig. 5b), the thunderstorm moved to the border between the mountains and the plain, and there was an evident V-shaped notch on its rear side, indicating a robust rear inflow. The area of the strong echo above 40 dBZ expanded further, with the echo center intensity peaking at more than 50 dBZ. It was important to note that during its propagation, the thunderstorm system broke down and gradually split into larger thunderstorm cell I and smaller thunderstorm cell II in the northwest corner of the built-up area, resulting in a significant barrier effect. Due to the barrier effect, the thunderstorm system began to split, with cell I and cell II remaining connected by a cloud bridge. At 21:06 BJT (Fig. 5c–d), the echo core volume of cell I decreased, and the cloud bridge began to narrow. The echo core height of cell II expanded to 8 km, with a maximum reflectivity exceeding 65 dBZ. Furthermore, newborn cells emerged around the cloud bridge. By 21:12 BJT (Fig. 5e–f), as the thunderstorm system approached the built-up area, the echo core of cell I and the cloud bridge weakened to 45 dBZ, while the echo core height and the strong echo area of cell II continued to increase. At 21:18 BJT (Fig. 5g–h), cell I and cell II were completely separated when the cloud bridge broke.

Through observational data, in this study, we analyzed in detail the variation of the thermal-dynamic field of the "0713" case passing over the built-up area. At 08:00 BJT (Beijing time), the wind direction below 850 hPa in the built-up area varied clockwise with height (red box in Fig. 6a), indicating a small amplitude and weak warm advection in the lower atmosphere. At this time, the urban heat island (UHI) intensity was 0.5 °C and the pseudo-equivalent temperature between 850 hPa and 925 hPa was $-10.9$ °C (purple column in Fig. 6d), indicating unstable atmospheric stratification in the lower atmosphere. In Fig. 6b and c, the specific humidity at 850 hPa exceeded $20 \, \mathrm{g \, kg^{-1}}$, indicating a rich water vapor content. Specific humidity at 500 hPa was approximately $4 \, \mathrm{g \, kg^{-1}}$, demonstrating the characteristics of the upper dry and lower wet vertical water vapor layers. The convective available potential energy (CAPE) value reached $3783.5 \, \mathrm{J \, kg^{-1}}$. At 14:00 BJT, the wind field in the middle and upper levels changed counterclockwise with height (shown in the blue box in Fig. 6a), representing a strong cold advection passing over the city. As the thunderstorm system moved into the built-up area, the configuration of upper cold and lower warm increased atmospheric stratification instability. Although the UHI intensity in the built-up area was only 0.8 °C (Fig. 6d), the pseudo-equivalent temperature between 850 and 925 hPa decreased to $-17.5$ °C, and the CAPE value exceeded $4000 \, \mathrm{J \, kg^{-1}}$. By 20:00 BJT, the UHI intensity, pseudo-equivalent temperature, and CAPE value continued to rise, leading to numerous CG lightning events initiated in the city periphery.

Figure 7a illustrates the movement trajectory of the "0713" case and the monitoring results of CG activities. At 19:00 BJT (Fig. 7b), the entire plain area of Beijing was dominated by a large-scale warm and humid south airflow. The warmest center was located within the built-up area, exhibiting a UHI intensity of approximately 1.2 °C. At 20:00 BJT (Fig. 7c), the squall line system originating from Taihang Mountain began to enter the northwest edge of the built-up area, creating a cold pool on the ground. The maximum wind speed at the front of this cold pool reached $7.1 \, \mathrm{m \, s^{-1}}$. A distinct convergence line was observed between the outflow boundary of the cold pool and the southerly winds of the environmental field in the west of the built-up area. This convergence line followed a northeast–southwest trajectory overall. The convergence zones triggered strong vertical upward movement of airflow in the lower layer, leading to 337 CG events in areas where the temperature gradient zones were most pronounced. At 21:00 BJT (Fig. 7d), as the thunderstorm system developed eastward, the cold pool area expanded further. The maximum wind speed at the front of the cold pool reached $10.4 \, \mathrm{m \, s^{-1}}$, and 417 CG events occurred around temperature gradient zones. It was worth noting that at this time the outflow boundary became bifurcated at the edge of the built-up area, and the convergence line began to fragment. At 22:00 BJT (Fig. 7e), the outflow angle of the cold pool expanded further. Under the influence of the bifurcated airflow, two cold tongues emerged on the west side of the built-up area. The convergence line broke completely, leading to a significant barrier effect. At this time, SGLNET only recorded 54 CG flashes near the split convergence line. At 23:00 BJT (Fig. 7f), the thunderstorm cell continued to propagate in the built-up area, and SGLNET registered 181 CG events within the built-up area.

Based on the above analysis, when the thunderstorm passed over the built-up area, it exhibited a bifurcated process due to the barrier effect. Utilizing this pattern as a screening criterion, we categorized thunderstorms passing over the built-up area of Beijing from 2010 to 2017 into bifurcated thunderstorms (BTs) and non-bifurcated thunderstorms (NBTs). According to Fig. S3 in the Supplement, the year with the highest number of BTs was 2013, with eight events, accounting for 23.5 % of the total thunderstorms; the lowest number of BTs was observed in 2010, with two events, representing 15.4 % of the total thunderstorms. These results indicated that the barrier effect of the urban underlying surface was a prevalent phenomenon in long-term thunderstorm observations. In summary, the urban underlying surface may be an important factor influencing thunderstorm bifurcation. Furthermore, the evolution of near-surface cold pools and convergence lines could serve as diagnostic indicators to understand how the urban underlying surface affected the thunderstorm process and CG activities.

## 3.3 Numerical simulation of the influence of urban underlying surface on the thermal-dynamic structure of thunderstorms

This section made a comparison between the observed values of AWS around the built-up area and the simulated values

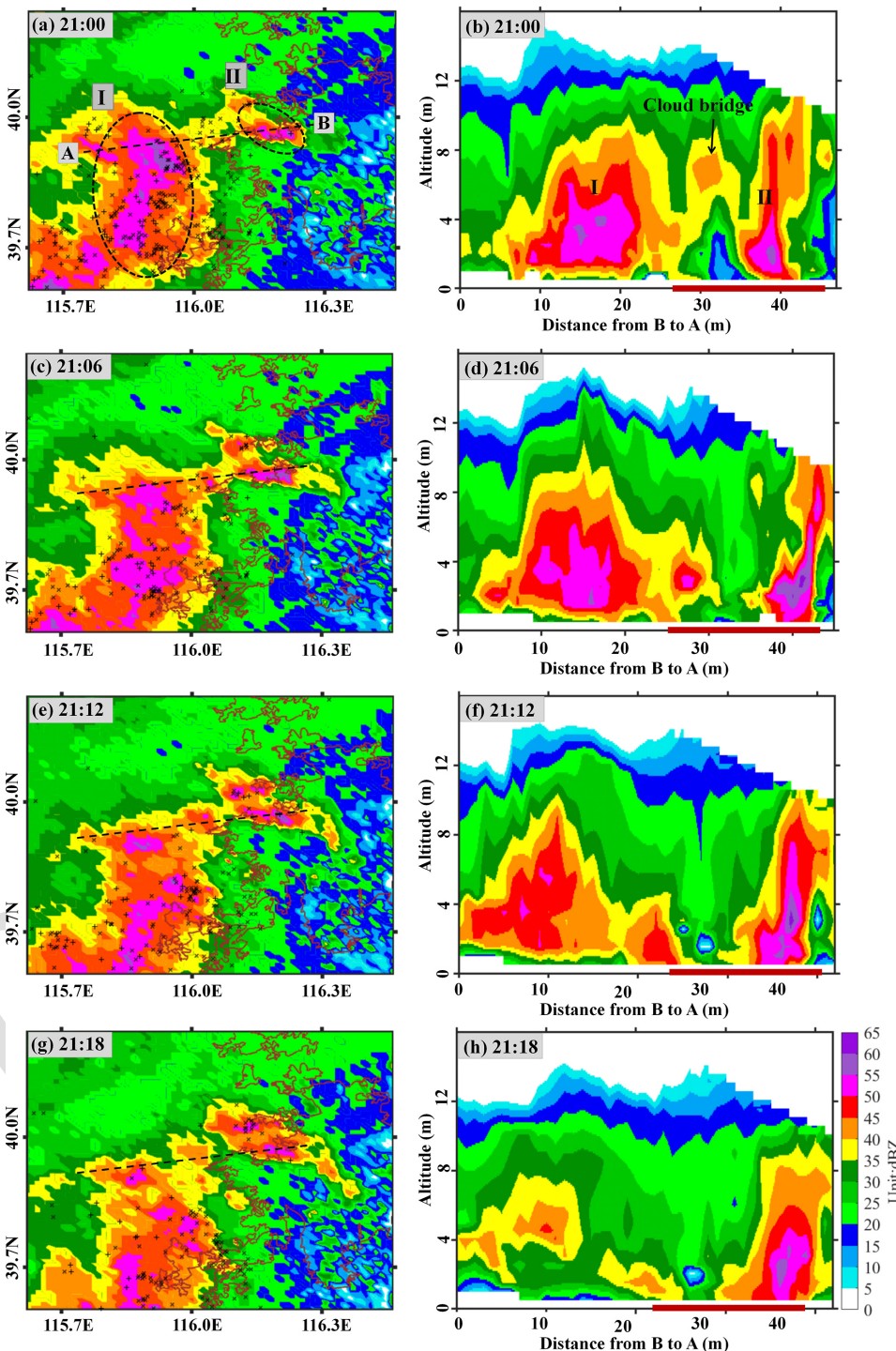

**Figure 5.** Temporal evolution of CR products for the "0713" thunderstorm case during its bifurcation stage: **(a, c, e, g)** horizontal reflectivity distributions at 21:00, 21:06, 21:12, and 21:18 BJT, respectively, and the **(b, d, f, h)** corresponding vertical cross sections along the AB line. TS3

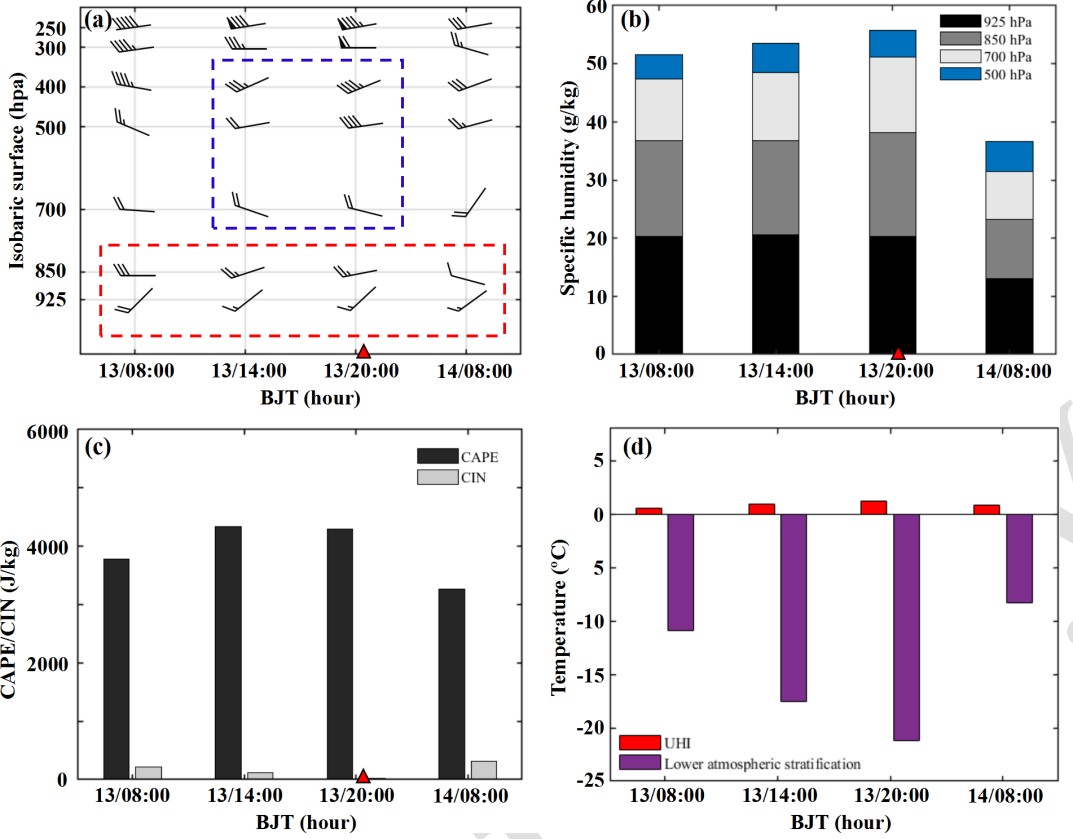

**Figure 6.** Wind field (**a**), specific humidity (**b**), convective effective potential energy (CAPE) and convective inhibition energy (CIN) (**c**), and urban heat island (UHI) intensity and lower atmospheric stratification (**d**). The red triangle represents the time when the "0713" case passed over the built-up area.

derived from the control experiments to verify the simulation accuracy.

Figure S4 in the Supplement illustrates the simulation accuracy of hourly meteorological elements. As the thunderstorm system progressed towards urban areas (specifically at 20:00 BJT), AWS documented a decrease in air temperature accompanied by an increase in air pressure. At 22:00 BJT, there was a notable increase in air pressure, a sharp drop in air temperature, and a concurrent rapid increase in wind speed. The simulated pressure exhibited a trend similar to that of the observed pressure, with the simulated pressure being 2.1 hPa lower than the observed pressure at 17:00 BJT. The curve of the simulated temperature showed a slight lag compared to that of the observed temperature. The variation in simulated wind speed closely mirrored that of the observed temperature, with both showing a rapid increase after 20:00 BJT. The emergence of these simulation errors could potentially be attributed to the chosen physical scheme, along with the initial and boundary conditions that were imported (Weisman and Davis, 1998; Jankov et al., 2007; Xu et al., 2013; Zheng et al., 2016). Compared to previous research, the simulation errors in our study were within an acceptable range. Overall, the simulated ground meteorological fields were able to

adequately capture the near-surface thermodynamic characteristics associated with the "0713" case.

Referring to previous research (Takemi, 2006; Yuan, 2015), this paper used a criterion of a perturbation potential temperature lower than $-1\,\mathrm{K}$ to define the core of the cold pool. The density flow outside this core, which spans a range of $-1$ to $1\,\mathrm{K}$, was also defined. In the experiment 1 (Fig. 8a), the simulation results showed that at 22:30 BJT, as the dry cold air sank and spread, the intensity of the near-surface cold pool increased. The minimum perturbation potential temperature of the cold pool core reached $-3\,\mathrm{K}$. It became evident that as the cold pool approached the edge of the built-up area, the outflow on both sides of the city moved faster, representing that the urban underlying surface dragged the cold pool. A cross section along the line CD revealed that due to the barrier effect of the underlying surface, the cold pool accumulated at the city periphery, with a core thickness of approximately $0.5\,\mathrm{km}$. The urban underlying surface altered the motion of the cold pool, making it unable to maintain a smooth and continuous shape, and then it bifurcated. A cross section along line EF revealed that the cold pool was divided into three relatively independent cores, with a more significant thickness at the edge of the city compared to its

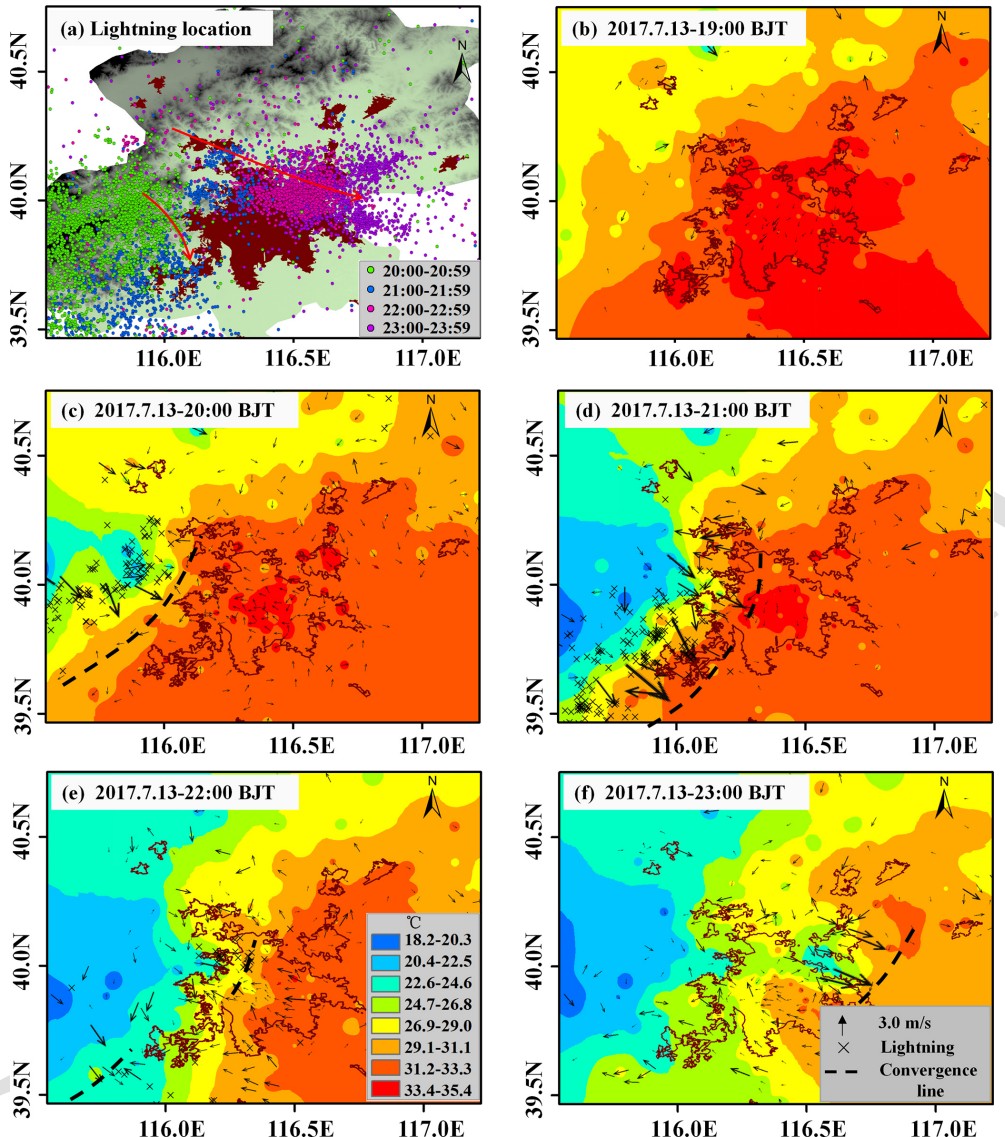

**Figure 7. (a)** Spatial distribution of CG lightning (color-coded by time intervals) with the thunderstorm movement trajectory (red line) overlaid. **(b–f)** Temporal evolution of near-surface thermodynamic (temperature) and dynamic (wind field, convergence lines) fields during the passage of the "0713" thunderstorm over the built-up area, at 19:00, 20:00, 21:00, 22:00, and 23:00 BJT, respectively.

inner regions. The length and area of the built-up area in Beijing are approximately 98.6 km and 600 km², respectively. In the experiment 2 (Fig. 8b), buildings outside the 5th Ring Road were replaced with bare land, which weakened the accumulation of the cold pool, resulting in a thickness of less than 0.4 km. This led to an increase in vertical speed over the city center, mitigating the separation of the cold pool. When all built-up areas were replaced with bare land in experiment 3 (Fig. 8c), the accumulation of cold pool cores disappeared. The cold pool cores exhibited a complete and continuous shape, with vertical upward movement continuing to increase, and the barrier effect almost disappeared.

In EXP1 (Fig. 9a), the horizontal field showed that the maximum wind speed at the front of the cold pool outflow reached approximately $5\,\mathrm{m\,s^{-1}}$. Here, the outflow boundary of the cold pool converged with the southerly winds of the environmental field, while divergence zones formed at the city center. The gust front at the exit of the cold pool exhibited bifurcation, with one segment moving southward and the other northward, leaving a sparse outflow in the middle. The cross section illustrated that the vertical ascending zone above the city center (denoted by the black dotted rectangle) aligned with the middle segment of the cold pool outflow, though its range and intensity were smaller compared to those on the city's periphery. In contrast, under the conditions

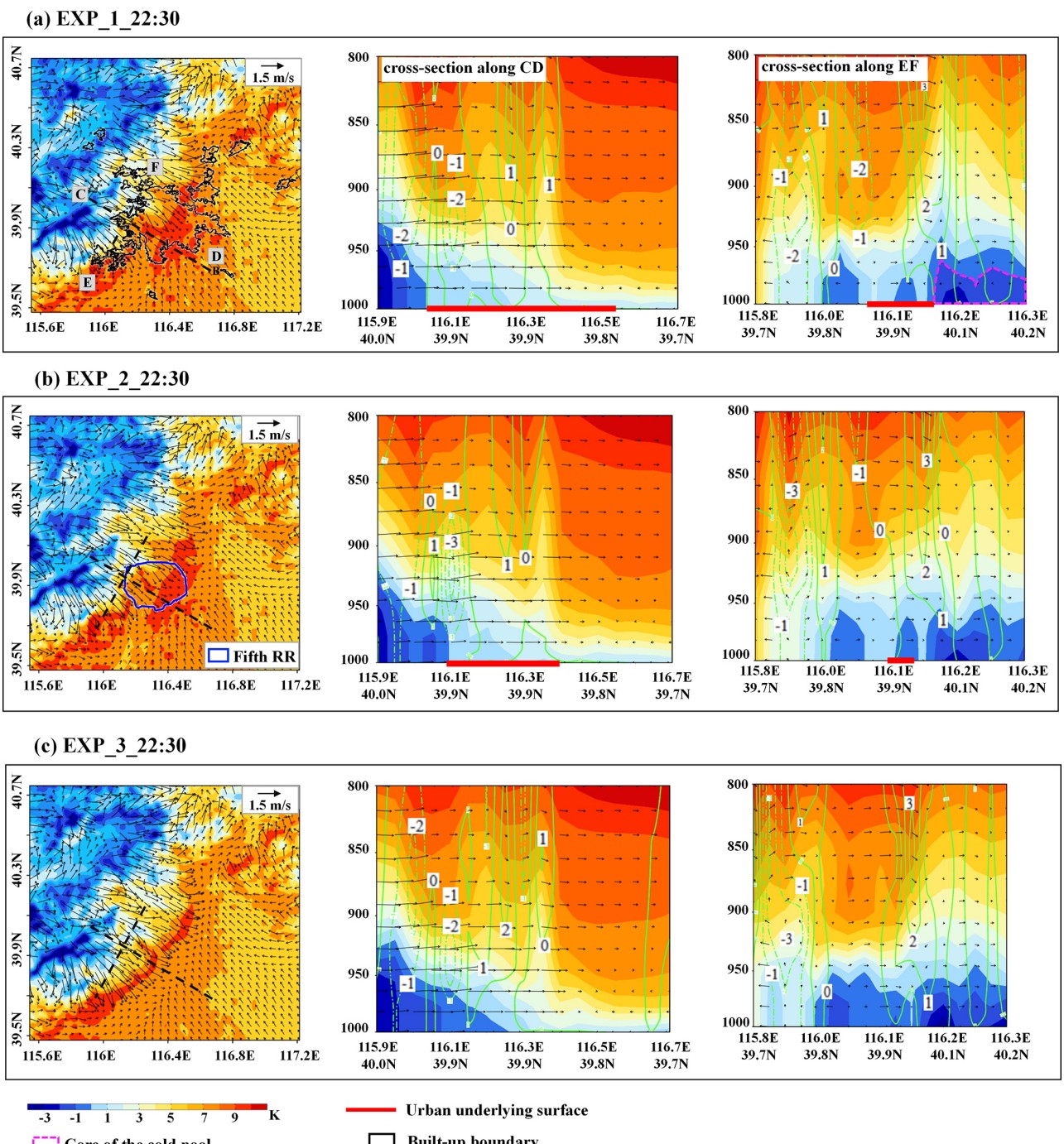

**Figure 8.** Influence of the scale of the built-up area on near-surface cold pool outflow simulated by WRF. Panels **(a)**–**(c)** represent experiments 1 to 3, respectively. In each set of subfigures, the left subfigure represents the horizontal thermal-dynamical field, and the middle and right subfigures depict the cross sections along the lines CD and EF, respectively, including perturbation potential temperature (shaded, unit: K), east–west circulation (vector), and vertical velocity (isoline, $\mathrm{m\,s^{-1}}$).

of EXP2 (Fig. 9b), where the roughness of the urban underlying surface was diminished, the cold pool outflow exhibited a slight bifurcation. This resulted in relatively intact convergence zones within the built-up area. The vertical ascending zone over the city center expanded in range, with a maximum

vertical velocity reaching about $4\,\mathrm{m\,s^{-1}}$. In EXP3 (Fig. 9c), the maximum wind speed at the front of the cold pool outflow surpassed $8\,\mathrm{m\,s^{-1}}$. The cold pool outflow completely traversed the built-up area, leading to a further enhancement of the range and intensity of the convergence zones, with a

vertical velocity of $5\,\mathrm{m\,s^{-1}}$. The above results showed that the scale of the built-up area could directly alter the strength of the barrier effect. Next, this section further investigated the impact of building density on the barrier effect. In EXP4 (Fig. 9d), all types of buildings were designated as open rise. Examination of the simulated dynamic field revealed that the bifurcation angle, the structure of the convergence zone, and the vertical velocity of the cold pool outflow were largely consistent with those observed in EXP1. In EXP5 (Fig. 9e), all types of buildings were set as compact rise. In particular, the separation angle of the cold pool outflow was larger than in EXP1, and the vertical velocity was smaller, indicating a more significant barrier effect. Therefore, it became evident that when a summer thunderstorm passed over the city, the specific morphological characteristics of the underlying surface could significantly modulate the thunderstorm process.

## 4 Discussion

Current understanding of urban surface impacts on lightning remains incomplete. Previous research has focused mainly on precipitation distribution (Dou et al., 2015; Qian et al., 2022; Yang et al., 2024), with Dou et al. (2024) proposing mechanisms for urban rainfall anomalies through thermal-dynamic processes. However, since precipitation peaks typically lag lightning activity (Li et al., 2017; Wu et al., 2018), these findings have limited direct applicability to lightning studies. Our research examines urban surface configurations, developing a conceptual model for urban lightning anomalies.

As Fig. 10a revealed, when thunderstorms passed over the small-scale city dominated by open rise, the barrier effect of the urban underlying surface on the dynamic field was constrained. The horizontal temperature gradient between the cold pool outflow and the underlying surface generated vertical shear of wind speed (Weisman and Klemp, 1984; Chen and Wang, 2012). This resulted in increased vertical airflow velocity, ensuring the continuation of ascending airflow (Knaff et al., 2004). Furthermore, the thermal circulation activated by the UHI effect contributed to the sustained development and organization of the thunderstorm system. Consequently, when the thunderstorms encountered the small-scale open rise, the thermal effect enhanced the thunderstorm process, and CG events were predominantly concentrated within the built-up area.

As illustrated in Fig. 10b, when thunderstorms passed over the large-scale city dominated by compact rise, the rough underlying surface caused significant attenuation of the horizontal dynamic field (Jin and Shepherd, 2005; Hand and Shepherd, 2009). The urban underlying surface altered the motion state of the cold pool, leading to stagnation and accumulation at the edge of the built-up area. Once the city size and building density reach a critical threshold, the outflow of the cold pool undergoes separation, resulting in the breakdown of the convergence line. Consequently, the vertical airflow velocity above the built-up area was significantly weakened, preventing the formation of new convective cells. The propagation speed of the thunderstorm cells above the built-up area was slower than on both sides of the city, resulting in bifurcation and movement around the thunderstorm system. Certainly, as the building density increased, the barrier effect became more pronounced. Therefore, when the thunderstorm system passed over the large-scale compact rise, the barrier effect of the urban underlying surface dominated the organization process of the thunderstorm, leading to the concentrated CG events in the city periphery.

## 5 Conclusion

Using observation data and numerical simulations, this article conducted a detailed investigation regarding how the urban underlying surface modulates thunderstorm processes and CG activities. We first analyzed the patterns of CG activities around the Beijing megacity over eight years. Subsequently, a thunderstorm case that passed over the built-up area of Beijing megacity was selected as a representative case to investigate the potential influence of the urban underlying surface on thunderstorm processes and CG activities.

Under similar climate conditions, the long-term pattern of the CG activities showed that Beijing, with its extensive city size, exhibited a notable barrier effect, while Zhangjiakou, a smaller city surrounding Beijing, did not exhibit a barrier effect. Thereafter, we counted the CG number across various LCZs, discovering that the CG density in compact rise was higher than that in open rise. These findings suggested that city size and building density could be significant factors influencing CG activities in the built-up area. The "0713" thunderstorm passing over the Beijing megacity was selected as a representative case study. Radar echoes and AWS data revealed that the evolution of the near-surface cold pool and convergence line can provide diagnostic indicators for investigating the influence mechanisms of urban underlying surfaces on the organization process of the "0713" thunderstorm. Our simulation results further indicated that the urban underlying surface altered the motion state of the cold pool and weakened the vertical airflow within the built-up area. Notably, the strength of the barrier effect varied with city size and building density. Finally, this paper established a conceptual model that illustrated how the urban underlying surface modulated the evolution of thunderstorm processes and the pattern of CG activities.

Broadly, our findings have implications in providing an important theoretical foundation for the forecasting and early warning of urban thunderstorm disasters. In the future, we plan to conduct high-resolution observations of meteorological elements within the urban boundary layer, aiming to deeply analyze the variations in boundary layer structure and energy balance when thunderstorms pass over the cities. Fur-

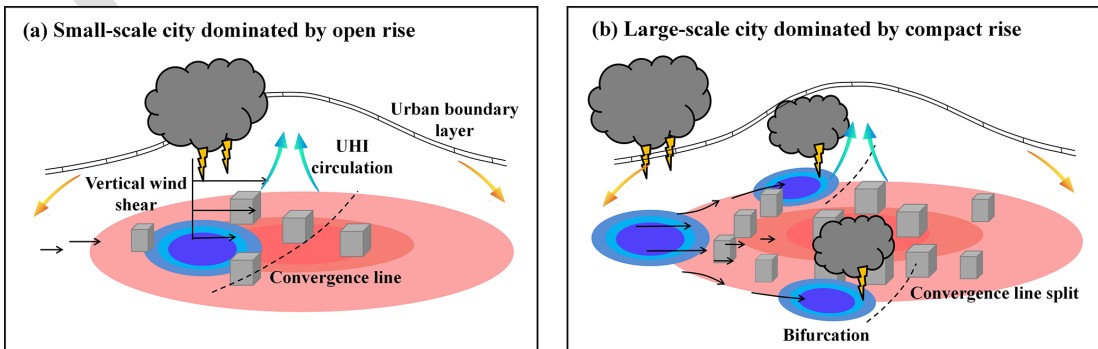

**Figure 9.** Influence of the urban underlying surface on near-surface convergence zones simulated by WRF. Panels **(a)**–**(e)** represent experiments 1–5, respectively. In each set of subfigures, the left subfigure represents the horizontal wind field and divergence field, while the right subfigure depicts the cross section of vertical velocity along the line GH.

**Figure 10.** Schematic diagram illustrating the modulation mechanisms of thunderstorm development processes and CG lightning activities by the urban configuration structure: a small-scale city dominated by open-rise construction **(a)** and a large-scale city dominated by compact-rise construction **(b)**.

thermore, we will design more detailed sensitivity experiments to comprehensively understand the mechanisms by which the urban underlying surface impacts thunderstorm disasters.

**Data availability.** Lightning detection data can be obtained upon request from the State Grid Electric Power Research Institute (http://www.sgepri.sgcc.com.cn TS4, last access: 2025). Urban form datasets are available upon request from the Institute of Urban Meteorology, China Meteorological Administration (https://www.ium.cn TS5, last access: 2025). Hourly AWS observation data can be obtained upon request from the China Meteorological Data Service Centre (http://data.cma.cn/en/ TS6, last access: 2025).

**Supplement.** The supplement related to this article is available online at [the link will be implemented upon publication].

**Author contributions.** TS, YY, and GL conceptualized the study. ZZ and YY performed the model development and conducted the simulations. TS wrote the original manuscript and plotted all the figures. YZ, YT, LL, and SL assisted in the conceptualization and model development. All the authors contributed to the manuscript preparation, discussion, and writing.

**Competing interests.** The contact author has declared that none of the authors has any competing interests.

**Disclaimer.** Publisher's note: Copernicus Publications remains neutral with regard to jurisdictional claims made in the text, published maps, institutional affiliations, or any other geographical representation in this paper. While Copernicus Publications makes every effort to include appropriate place names, the final responsibility lies with the authors.

**Acknowledgements.** The data that support the findings of this study are available from the Institute of Urban Meteorology, China Meteorological Administration, and the State Grid Electric Power Research Institute upon applying for cooperation. TS7

The authors express their gratitude to the editor and four anonymous reviewers for their constructive comments and suggestions to improve this paper.

**Financial support.** This research was supported by the National Natural Science Foundation of China (grant nos. 42222503, 42175098, and 42105147), the Fund of the Key Laboratory of Transportation Meteorology, China Meteorological Administration & Nanjing Joint Institute for Atmospheric Sciences (grant no. BJG202506), and the Fund of the Key Laboratory of Radar Meteorology of the China Meteorological Administration (Study on the thermodynamic and dynamic mechanisms of urban canopy's impact on summer thunderstorm activity).

**Review statement.** This paper was edited by Eduardo Landulfo and reviewed by four anonymous referees.

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

## Remarks from the typesetter

TS1    Please confirm update of affiliation.

TS2    Please note that we only abbreviate "Figure" to "Fig." in the middle of a sentence, not at the beginning.

TS3    Please note that the requested changes (especially changing the units in the figure) will require editor approval. Please provide a short explanation regarding the corrections to this figure that can be forwarded by us to the editor.

TS4    Please provide a direct link to the data set and, if possible, a DOI instead of a URL. In any case, please provide a reference list entry including creators, title, and date of last access.

TS5    Please provide a direct link to the data set and, if possible, a DOI instead of a URL. In any case, please provide a reference list entry including creators, title, and date of last access.

TS6    Please provide a direct link to the data set and, if possible, a DOI instead of a URL. In any case, please provide a reference list entry including creators, title, and date of last access.

TS7    Should this first part of the acknowledgements be deleted?

TS8    Please check DOI.