# Peer review of "Urban underlying surface modulates summertime thunderstorm processes and associated lightning activities"

_Atmospheric Chemistry and Physics, 2024_

## Author Comment (AC1)

**Response to Reviewer Comments of the First Reviewer**

Dear Reviewer and Editors:

We are sincerely grateful to the editor and reviewer for their valuable time for reviewing our manuscript. The comments are very helpful and valuable, and we have addressed the issues raised by the reviewer in the revised manuscript. Please find our point-by-point response (in blue text) to the comments (in black text) raised by the reviewer. We have revised the paper according to your comments (highlighted in red text of the revised manuscript).

**Overview:**

**This study analyzed the impact of urban areas on thunderstorm organization processes and CG flash activity through ground observations and numerical simulations. City size may be an important factor affecting thunderstorm processes. In addition, the building density may also alter the organization process of thunderstorms. Overall, I believe that the research presented in this article has some innovation and the conclusions are also very interesting. The organization and writing of this article need improvement, and I would like to suggest significant revisions to this paper.**

**Response:** Thank you for your recognition of our work and for your valuable feedback. As per your request, we have undertaken significant revisions throughout the manuscript.

Firstly, we have expanded the Introduction section to provide a more detailed overview of relevant research in the field. We refresh and highlight the scientific question on how to modulate thunderstorm disasters on the urban underlying surface of Beijing.

Secondly, in the Data and Methodology section, we have revised the descriptions of

lightning location data, radar products, and LCZ datasets. These updates ensure that readers have a clear understanding of the analytical approach we have taken.

Lastly, we have thoroughly revised the Abstract and Conclusion sections to highlight our main findings. Specifically, we have reorganized the mechanisms by which urban underlying surface play different dominant role in thunderstorm disasters.

We are grateful for your time and effort in reviewing our work and for your guidance in improving the manuscript. We hope that these revisions have addressed your concerns and have enhanced the clarity of our research.

**Major comments:**

**1. The author has failed to provide a definition of a thunderstorm that is grounded in radar reflectivity or other pertinent parameters, leaving the reader without a clear understanding of the term's scientific context.**

**Response:** I apologize for any confusion caused by the lack of clarity in defining thunderstorms based on radar echoes in our previous submission. This was indeed an oversight on our part, and we sincerely appreciate your bringing it to our attention.

In response, we have made substantial revisions in the Data section of the manuscript. Specifically, we have provided additional information on the radar data used and the clearer criteria for identifying thunderstorms from the radar echoes. Line 98-107 in the revised manuscript:

"Doppler Radar Data. This radar observation system consists of a data acquisition subsystem, a product generation subsystem, and a main user terminal subsystem. It enables real-time data transmission and image stitching, significantly boosting the monitoring and early warning capabilities for disastrous weather conditions such as severe convective weather, tropical cyclones, and heavy rainfall. The radar data employed in this paper is the Composite Reflectivity (CR) product generated by the S-band Doppler Radar stationed at the Beijing Nanjiao Observatory. Previous studies have consistently recognized a threshold of 35 dBZ as a pivotal marker signifying the presence of a convective echo (Dixon & Wiener, 1993; Roberts & Rutledge, 2003; Mecikalski & Bedka, 2006). Consequently, this research adopts this well-established

reflectivity threshold as the criterion for identifying thunderstorms. In addition, to gain a broader understanding of the synoptic background of thunderstorms, we utilized sounding data from the Beijing Nanjiao Observatory. These sounding data were collected at 02:00, 08:00, 14:00, and 20:00 Beijing time every day."

Reference:

Dixon M, Wiener G.: TITAN: thunderstorm identification, tracking, analysis, and nowcasting–A radar-based methodology. Journal of Atmospheric and Oceanic Technology, 10, 785–797, https://doi.org/10.1175/1520-0426(1993)0102.0.CO;2, 1993.

Mecikalski, J. R., Bedka, K. M.: Forecasting convective initiation by monitoring the evolution of moving cumulus in daytime GOES imagery. Monthly Weather Review, 134, 49–78, https://doi.org/10.1175/MWR3062.1, 2006.

Roberts, R. D., Rutledge, S.: Nowcasting storm initiation and growth using GOES-8 and WSR-88D data. Weather and Forecasting, 18, 562–584, https://doi.org/10.1175/1520-0434(2003)0182.0.CO;2, 2003.

**2. This article highlights a specific thunderstorm process that exhibited barrier effect through both observations and simulations. However, to strengthen the argument that this phenomenon is widespread or common, the author should provide additional cases or statistical results to support their findings.**

**Response:** Thank you for bringing to my attention the lack of clarity regarding the definition of thunderstorms. I apologize for any confusion caused by the previous omission. I have now supplemented the manuscript with the necessary information. Specifically, I have included statistical results from recent years, focusing on thunderstorms that have traversed the Beijing area and exhibited the barrier effect. Line 254-262 in the revised manuscript:

"Based on the above analysis, when the thunderstorm passed over the built-up area, it exhibited a bifurcated process due to the barrier effect. Utilizing this pattern as a screening criterion, we categorized thunderstorms passing over the built-up area of Beijing from 2010 to 2017 into bifurcated thunderstorms (BT) and non-bifurcated

thunderstorms (NBT). According to Figure S3, the year with the highest number of BT was 2013, with eight events, accounting for 23.5% of the total thunderstorms; the lowest number of BT was observed in 2010, with two events, representing 15.4% of the total thunderstorms. These results indicated that the barrier effect of the urban underlying surface was a prevalent phenomenon in long-term thunderstorm observations."

[Figure]

**Figure S3: Interannual variation of bifurcated thunderstorms (BT) and non-bifurcated thunderstorms (NBT) in the built-up area of Beijing.**

**3. Has Figure 3 exclusively analyzed CG events that occurred during the summer? If so, please include a relevant description in the caption of Figure 3. Furthermore, the author should provide a clear description of the data in the data section.**

**Response:** Thank you for bringing this to our attention. We apologize for the oversight and have made the necessary revisions to clarify the information.

We have updated the caption of Figure 3 to explicitly state that it exclusively analyzes CG events that occurred during the summer months. Additionally, in the Data section of our manuscript, we have included a clear description of the data used, specifying that the summer period refers to June through August.

We appreciate your careful review and hope these revisions address your concerns. Please let us know if you have any further questions or suggestions.

[Figure]

**Figure: 3 Overview of the built-up areas in Beijing (BJ), Zhagnjiakou (ZJK), and Tianjin (TJ) (a). Spatial patterns of CG density in the built-up areas of Beijing (b), Zhangjiakou (c), and Tianjin (d) during the summertime of 2010-2017.**

**4. Please add the symbolization of red line in Figure 8.**

**Response:** Thank you for bringing this to our attention. We sincerely apologize for any confusion caused by the absence of clear symbolization for the red line in Figure 7 (original Figure 8). In response to your feedback, we have promptly revised the figure to include a clear explanation of the meaning of the red line.

Furthermore, we have conducted a thorough review of all figures in the manuscript to prevent any similar oversights in the future.

We appreciate your careful scrutiny and the opportunity to improve our work.

[Figure]

**Figure 7: (a) The spatial pattern of CG activities. The dots represent cloud-to-ground lightning, and the red line represents the movement trajectory of the thunderstorm. (b-f) The near-surface thermal-dynamic fields of the "0713" case passing over the built-up area.**

---

## Author Comment (AC2)

**Response to Reviewer Comments of the Second Reviewer**

Dear Reviewer and Editors:

We are sincerely grateful to the editor and reviewer for their valuable time for reviewing our manuscript. The comments are very helpful and valuable, and we have addressed the issues raised by the reviewer in the revised manuscript. Please find our point-by-point response (in blue text) to the comments (in black text) raised by the reviewer. We have revised the paper according to your comments (highlighted in red text of the revised manuscript).

**This manuscript examines lightning location data, along with a numerical model, which indicates that urban morphology in Beijing's metropolitan area influence where lightning strikes一a phenomenon known as the urban barrier effect. By integrating lightning data with a model that considers building size and height, the manuscript presents interesting case studies demonstrating how large cities can influence weather conditions, thus affecting lightning patterns. However, the manuscript lacks sufficient clarity in its data presentation, methodological approach and structural hierarchy, causing confusion for readers. After addressing my primary comments, I recommend publishing this manuscript in ACP.**

**Response:** Thank you for your recognition of our work and for your valuable feedback. As per your request, we have undertaken significant revisions throughout the manuscript.

**Major comments:**

**1. Figure 1 provides an overview of the study area, encompassing the boundaries and spatial locations of China, Beijing-Tianjin-Hebei region, and Beijing.**

**However, as the focus of this study is the urban area of Beijing, Figure 1 lacks a sufficiently detailed depiction. While Figure 2 offers some insight, I believe it is crucial to supplement Figure 1 with more specific information about the urban area of Beijing.**

**Response:** Thank you for your valuable feedback. I appreciate your observation that while Figure 1 provides a broad overview of the study area, including China, the Beijing-Tianjin-Hebei region, and Beijing, it may lack sufficient detail for the specific focus of this study, which is the urban area of Beijing.

To address this, I agree that it is essential to enhance Figure 1 or introduce an additional figure that presents a more detailed depiction of Beijing's urban area. I have revised the figure to include a closer look at the topography and built-up area in Beijing megacity.

[Figure]

Figure 1: Overview of the study area (a). Topography and built-up area in Beijing megacity (b).

**2. The observation section of this article has obtained many interesting statistical results based on compact high rise, compact mid rise, compact low rise, open high rise, open mid rise, and open low rise. However, sensitivity tests were conducted in the simulation section based on compact rise and open rise, and the test results to some extent explained the mechanism of the influence of urban morphology on the process of thunderstorm organization. What are the standards used for the classification of urban morphology in the observation and**

**simulation sections? Is it consistent? Please provide a detailed explanation.**

**Response:** I apologize for any confusion my previous explanation may have caused regarding the classification of urban morphology in the observation and simulation sections of our article. Allow me to clarify and provide a more detailed explanation.

In the observation section, we classified urban morphology into six distinct categories: compact high-rise, compact mid-rise, compact low-rise, open high-rise, open mid-rise, and open low-rise. Please refer to Fig. S1 and Tab. S1 for a spatial morphology diagram and classification criteria. This comprehensive classification scheme was designed to capture the nuances in urban structures that might impact the thunderstorm process and CG activity.

For the simulation part (Fig. 2), the classification criteria for urban morphology are consistent with those for the observation part. However, considering the computational efficiency of the model, the simulation scheme of WRF focuses on two broad categories: compact rise and open rise. The compact rise includes compact high-rise, compact mid-rise, compact low-rise. The open rise includes open high-rise, open mid-rise, and open low-rise. This simplification helps us effectively explore the potential physical effects of urban morphology on thunderstorm organization processes while ensuring computational efficiency.

I hope this clarifies our classification schemes and addresses your concerns about consistency. Thank you for your patience and valuable feedback.

[Figure]

**Figure S1: Schematic diagram of urban morphology based on different categories of LCZ datasets.**

**Table S1: Descriptions and attribute parameters of different categories of LCZ datasets.**

| LCZ datasets | Descriptions | Attribute parameters |
|---|---|---|
| LCZ 1: Compact high-rise | The building are taller than 10 stories. High density of buildings and ground cover mostly hard pavement with little vegetation. | Aspect ratio >2
Sky view factor: 0.2-0.4
Building surface fraction: 40-60
Impervious surface fraction: 40-60
Pervious surface fraction < 10
Height of roughness elements >25 |
| LCZ 2: Compact mid-rise | Building heights span from 3 to 9 stories. High density of buildings of buildings and ground cover mostly hard pavement with little vegetation. | Aspect ratio: 0.75-2
Sky View Factor: 0.3-0.6
Building surface fraction: 40-70
Impervious surface fraction: 30-50
Pervious surface fraction <20
Height of roughness elements: 10-25 |
| LCZ 3: Compact low-rise | Building heights ranging from 1 to 3 stories. High density of buildings and ground cover mostly hard pavement with little vegetation. | Aspect ratio: 0.75-1.5
Sky view factor: 0.2-0.6
Building surface fraction: 40-70
Impervious surface fraction: 20-50
Pervious surface fraction <30
Height of roughness elements: 3-10 |
| LCZ 4: Open High-rise | Building heights of 10 stories or more. Low density of buildings and low ground cover mostly permeable ground or vegetation. | Aspect ratio: 0.75-1.25
Sky View Factor: 0.5-0.7
Building surface fraction: 20-40
Impervious surface fraction: 30-40
Pervious surface fraction: 30-40
Height of roughness elements >25 |
| LCZ 5: Open mid-rise | Building heights in the range of 3-10. Low density of buildings and low ground cover mostly permeable ground or vegetation. | Aspect ratio: 0.3-0.75
Sky view factor: 0.5-0.8
Building surface fraction: 20-40
Impervious surface fraction: 30-50
Pervious surface fraction: 20-40
Height of roughness elements: 10-25 |
| LCZ 6: Open low-rise | Building heights ranging from 1 to 3 stories. The low density of buildings and the ground cover is mostly permeable ground or vegetation. | Aspect ratio: 0.3-0.75
Sky View Factor: 0.6-0.9
Building surface fraction: 20-40
Impervious surface fraction: 20-50
Pervious surface fraction: 30-60
Height of roughness elements: 3-10 |

[Figure]

**Figure 2: Terrain height distribution and the building types of the WRF mesoscale numerical model.**

**3. Weather Research and Forecasting (WRF) serves as a vital tool in this paper for analyzing the mechanisms of how urban morphology impacts the thunderstorm process. However, the author's introduction to WRF lacks sufficient detail, especially regarding the WRF model coupled with the urban canopy, which poses challenges for researchers unfamiliar with this system. The author should provide a more comprehensive explanation of WRF and the simulation scheme to enhance the readability of the paper and ensure the reproducibility of the experimental results.**

**Response:** Thank you for your insightful comments on our manuscript. We apologize for any lack of clarity in our discussion of the WRF model and its integration with the urban canopy model. We understand that this may have been confusing for readers who are less familiar with these modeling techniques.

In response to your comments, we have added an additional detailed explanation regarding the WRF model and its integration with the urban canopy model in line 10-15. This enhanced description aims to provide a more comprehensive understanding of the model's capabilities.

We hope that these revisions will improve the readability of our paper. We are grateful for your feedback and the opportunity to strengthen our work through these revisions. Line 114-135.

[revised manuscript text omitted]

**Minor comments:**

**P1Line15: "mega cities", should be "megacities"**

**Response:** We greatly appreciate the time and patience you have taken to provide your insights on my work. I deeply apologize for the small errors and oversights that you have pointed out.

I have carefully addressed each of your minor comments and double-checked the entire manuscript for any other potential issues.

Once again, I apologize for any inconvenience caused by these minor errors and appreciate your patience and understanding.

**P1Line18: "weakening" should be preceded by "the"**
**Response:** Amended and thanks.

**P1Line30: "population" should be revised as "populations"**
**Response:** Amended and thanks.

**P2Line45: "that" in "....despite that....." should be deleted**
**Response:** Amended and thanks.

**P2Line49: "millions"  ->  "million"**
**Response:** Amended and thanks.

**P2Line61: "built up"  ->  "built-up"**
**Response:** Amended and thanks.

**P2Line67: What does "BJ" mean? Is Beijing an abbreviation?**
**Response:** Amended and thanks.

**P2Line67: "sof" may be "of"**

**Response:** Amended and thanks.

**P4Line96: "refers"   ->   "refer"**

**Response:** Amended and thanks.

**P6Line118: deleted "the" in "....the Table 2....."**

**Response:** Amended and thanks.

**P6Line125: deleted "the" in "....the Table 3....."**

**Response:** Amended and thanks.

**P8Line141: "an average and maximum densities"**

**->   "an average and maximum density"**

**Response:** Amended and thanks.

**P8Line145: "exists   a potential" has one more space**

**Response:** Amended and thanks.

**P9Line146: "by high" should be added "a"**

**Response:** Amended and thanks.

**P12Line203: "during "0713" case" should be added "the"**

**Response:** Amended and thanks.

**P18Line305: "all types of building" should be "all types of buildings"**

**Response:** Amended and thanks.

---

## Author Response (AR3)

**Response to Reviewer Comments**

Dear Reviewer and Editors:

We are sincerely grateful to the editor and reviewer for their valuable time for reviewing our manuscript. The comments are very helpful and valuable, and we have addressed the issues raised by the reviewer in the revised manuscript. Please find our point-by-point response (in blue text) to the comments (in black text) raised by the reviewer. We have revised the paper according to your comments (highlighted in red text of the revised manuscript).

**This paper studies the formation of cloud-to-ground lightning events using observations and numerical simulations in Beijing, China. First, they used SGLNET data to relate the frequency of CG events to built-up areas in three locations in China. Then, they analyze Beijing in more detail, considering the type of buildings and LCZs in relation to CG event density. Finally, they analyze the evolution of one day in more detail, first with observations and then with numerical simulations. They use WRF integrated with UCM to represent 5 experiments of varying land use. The paper is interesting, but some aspects need to be improved to understand its scientific contribution and deliver its insights to readers more clearly.**

**Response:** Thank you for your recognition of our work and for your valuable feedback. As per your request, we have undertaken significant revisions throughout the manuscript.

**Major comments:**

**1. While the introduction gives a detailed state of the art, it also lacks details regarding the methods used in previous studies. The reader is left alone in figuring out if the simulation setup is novel, if only the case study has never been**

**considered before, or if the datasets are unique.**

**Response:** We apologize for the lack of detail regarding the methods used in previous studies in our introduction. We acknowledge your feedback and have revised the introduction to better highlight the innovations of our study in utilizing numerical simulations to investigate the impact of the urban underlying surface on thunderstorm processes and lightning activities.

Scholars have explored the dynamic effects of urban underlying surfaces through numerical simulations. The results from global and regional climate models show that the urban rough underlying surface can alter the horizontal wind field, enhancing convergence and upward movement in the upstream direction (Jin & Shepherd, 2005), which, to some extent, facilitates the development of thunderstorm systems (Yin et al., 2020). The "climbing" upward airflow movement, as simulated by the WRF model (Zhu et al., 2016), exhibits a relatively weak intensity, insufficient to significantly alter the organizational processes of thunderstorms. Moreover, through urban boundary layer model simulations, researcher have also discovered that when thunderstorms pass over cities, the dynamic effect of the urban underlying surface can lead to the bifurcation and movement around of thunderstorm systems (Bernstein & LeRoy, 1990). This phenomenon is known as the barrier effect (Stallins & Bentley, 2006). The aforementioned simulation work has made valuable explorations into studying the dynamic effects of urban underlying surfaces. However, the current research only employs simulation scenarios with and without an urban underlying surface, without delving into the detailed characteristics of urban spatial configuration. It is worth noting that, within the spatial configuration of the urban underlying surface, city size is recognized as a crucial factor impacting thunderstorm processes (Kingfield et al., 2018, as shown in Fig. R1). Additionally, building density also demonstrates a tendency to alter urban lightning activities (Stallins & Bentley, 2006, as illustrated in Fig. R2). Therefore, it is necessary to comprehensively consider the spatial configuration characteristics of the urban underlying surface and continue to explore the influence mechanisms of the urban underlying surface on thunderstorm processes and associated lightning activities.

Thank you for your valuable comments, which have helped us to improve our manuscript.

[Figure]

**Figure R1: Storm tracks identified using the criteria specified in section 4b. The total number of storms occurring around each city along with the number of storms that started upwind and ended over the city (AB), started over the city and ended downwind (BC), and crossed all three regions (ABC) are annotated in black for the full climatology and UF datasets. The prevailing directions of motion for thunderstorm objects from both datasets are annotated in purple. (Kingfield et al., 2018)**

[Figure]

**Figure R2: Flash production and high-density urban land use (a), and flash production and low-density urban land use (b). (Stallins & Bentley, 2006)**

**Reference:**

Bornstein, R., and LeRoy M.: Urban barrier effect on convective and frontal thunderstorms. Preprints, Fourth Conf. on Mesoscale Processes, Boulder, CO, American Meteorological Society, 120–121, 1990.

Jin, M. L., Shepherd, J. M.: Inclusion of urban lands CAPE in a climate model: how can satellite data help?, Bulletin of the American Meteorological Society, 86, 681–689, https://doi.org/10.1175/BAMS-86-5-681, 2005.

Kingfield, D. M., Calhoun, K. M., de Beurs, K. M., Henebry, G. M.: Effects of city size on thunderstorm evolution revealed through a multiradar climatology of the central United States, Journal of Applied Meteorology and Climatology, 57, 295–317, https://doi.org/10.1175/JAMC-D-16-0341.1, 2018.

Stallins, J. A., and Bentley, M. L.: Urban lightning climatology and GIS: An analytical framework from the case study of Atlanta, Georgia, Applied Geography, 26, 3–4, 242–259, https://doi.org/10.1016/j.apgeog.2006.09.008, 2006.

Yin, J., Zhang, D., Luo, Y., and Ma, R.: On the Extreme Rainfall Event of May 7th 2017 Over the Coastal City of Guangzhou. Part I: Impacts of Urbanization and Orography, Monthly Weather Review, 148, 3, 955–979, https://doi.org/10.1175/MWR-D-19-0212.1, 2020.

Zhu, Y., Liu, H., and Shen, J.: Influence of Urban Heat Island on Pollution Diffusion in Suzhou, Plateau Meteorology, 35, 6, 1584–1594, https://doi.org/10.7522/j.issn.1000-0534.2016.00084, 2016.

**2. The numerical simulation is interesting, and some main features are given, but it is unclear how detailed the different descriptions of buildings/terrain are in the physical model. How fine is that in relation to the resolution of the simulation? Also, they mention that there are acceptable errors between the simulation and observations, but they should be addressed more carefully. Can sensitivity tests support the modeling effort to better understand its performance? This could be particularly interesting in the land use setup. Also, I'm guessing this was not explored, but there are some works about lightning models in WRF; was that an option here?**

**Response:** Many thanks for your kind suggestions, which were favored for this paper to be more rigorous and professional. The three questions raised in overall review were answered as followings:

**Comments 1: Regarding the resolution of the buildings/terrain in the physical model**

**Response:** We apologize for any confusion our previous presentation may have caused.

In this study, we utilized the WRF 4.0 version, integrated with UCM, configured with a triple-nested grid system. The horizontal resolutions of the nested grids are 5 km (D01), 1 km (D02), and 200 m (D03), respectively. Regarding the detail level of building and terrain descriptions, we employed a multi-resolution strategy to optimize computational resources and simulation accuracy. In non-critical areas, such as suburban regions, we adopted the lower-resolution terrain descriptions provided by the system to conserve computational resources and enhance simulation efficiency.

In critical areas, specifically urban zones, we incorporated high-precision three-dimensional building models and data. The underlying surface data used in our model encompasses land use and urban canopy datasets with a resolution of 10 m. These detailed models provide an accurate representation of urban features, such as building height, shape, and spatial distribution, which are crucial for capturing urban-induced circulations and other microclimate phenomena.

**Comments 2: Regarding the errors between the simulation and observations**

**Response:** Thank you for highlighting the significant issue of simulation errors in our paper. You are correct in pointing out that a deeper analysis is necessary given the numerical simulation conducted.

Actually, deviations between simulated results and actual observations are indeed a common phenomenon (Shimada et al., 2011; Pedro et al., 2011; Zhang et al., 2012; Zheng et al., 2016). Scholars used the WRF model to simulate a squall line event and found a 2-3 hour lag in the simulated daily maximum temperature at lower levels compared to actual observations, as well as a 2-3°C underestimation in the simulated temperature decrease in the evening. Compared to previous research, our simulated temperature curves lagged behind actual observations by about 1 hour and were 1-2°C lower than actual observations. Overall, the simulation errors in our study were within an acceptable range and accurately reflected the near-surface thermodynamic field

characteristics during the thunderstorm's passage over the city.

Regarding the causes of simulation errors, the primary reason for deviations between simulated results and actual observations lies in the coarse spatiotemporal resolution of conventional observational data and reanalysis data. This makes it difficult to accurately capture meteorological element information of mesoscale convective systems, leading to discrepancies in initial and boundary conditions compared to actual observations (Klemp, 1987; Weisman & Davis, 1998). Although initial errors may be small, they accumulate over time, ultimately resulting in significant deviations from actual observations.

Furthermore, the simulation capabilities of the model itself are also important factors contributing to errors. Due to limitations in model resolution, it is not possible to accurately describe changes in meteorological elements for every air parcel, or our understanding of atmospheric physical processes may be incomplete. Therefore, parameterization schemes such as those for the boundary layer, microphysical processes, and land surface processes are introduced into the model. These parameterization schemes, as approximations of atmospheric physical processes, also introduce certain simulation errors (Gallus & Bresch, 2006; Jankov et al., 2007). Although adjusting parameterization schemes can improve simulation results to some extent, due to the sensitivity of different weather process simulations to parameterization schemes, there are limitations in improving simulation accuracy through this method.

Of course, the simulation errors you mentioned are indeed unavoidable issues. In the future, we will continue to delve into the sources and influencing factors of simulation errors and explore more effective error control and improvement methods to enhance the accuracy and reliability of numerical simulations. Thank you again for your constructive comments.

**Comments 3: Regarding the lightning models in WRF**

**Response:** Thank you very much for your excellent suggestion on incorporating the Weather Research and Forecasting (WRF) model with lightning considerations.

The E-WRF system, primarily developed by the NOAA National Severe Storms

Laboratory (NSSL), has been implemented within the mesoscale numerical model WRF by Fierro et al. (2013). E-WRF offers a more explicit representation of electrical processes and lightning activity in mixed-phase convective clouds by coupling an explicit charging scheme and lightning parameterization into the NSSL 2-moment 4-ice microphysics scheme (Ziegler, 1985; Mansell et al., 2010; Mansell & Ziegler, 2013). A recent study by Lu (2022) utilized E-WRF to investigate the microphysical and electrical processes of a supercell squall line in Beijing. In Fig. R3, although some discrepancies were observed between the simulated radar echoes and lightning compared to observations, the overall simulation of lightning activity, both temporally and spatially, was deemed reliable and supportive of further in-depth analysis of macro- and micro-scale characteristics during thunderstorm development.

The focus of our current paper lies in analyzing the impact of urban underlying surfaces on thunderstorm processes and lightning activity. While the multi-layer canopy model coupled with WRF (WRF-UCM) cannot directly simulate lightning, it excels in fine-tuning the simulation of various spatial configurations of urban underlying surfaces, thereby influencing urban boundary layer conditions, thermodynamic structures during thunderstorm occurrences, and radar echo characteristics. This is an area where E-WRF currently falls short compared to WRF-UCM.

Of course, in the future, we will endeavor to couple the UCM with the E-WRF to investigate the direct impact of the complex geometric information of cities in the vertical direction on thunderstorm electrical processes and lightning activity. Such an integration would provide a more holistic understanding of urban-induced modifications on thunderstorm dynamics and associated hazards. Thank you again for your valuable insights, which have enriched the perspective of our research endeavors.

[Figure]

**Figure R3: The normalized total lightning frequency from the Observed results of (depicted in red, corresponding to the bottom red timeline) and simulated results from the E-WRF (depicted in blue, corresponding to the top blue timeline) (Lu, 2022).**

**Reference:**

Fierro, A. O., Mansell, E. R., MacGorman, D. R., et al.: The Implementation of an Explicit Charging and Discharge Lightning Scheme within the WRF-ARW Model: Benchmark Simulations of a Continental Squall Line, a Tropical Cyclone, and a Winter Storm, Monthly Weather Review, 141, 7, 2390–2415, http://doi.org/10.1175/MWR-D-12-00278.1, 2013.

Mansell, E. R., Ziegler, C. L., Bruning, E. C.: Simulated Electrification of a Small Thunderstorm with Two-Moment Bulk Microphysics, Journal of the Atmospheric Sciences, 67, 1, 171–194, http://doi.org/10.1175/2009JAS2965.1, 2010.

Mansell, E. R., Ziegler, C. L.: Aerosol Effects on Simulated Storm Electrification and Precipitation in a Two-Moment Bulk Microphysics Model, Journal of the Atmospheric Sciences, 70, 7, 2032–2050, http://doi.org/10.1175/JAS-D-12-0264.1, 2013.

Ziegler, C.: Retrieval of Thermal and Microphysical Variables in Observed Convective Storms. 1. Model Development and Preliminary Testing, Journal of the Atmospheric Sciences, 42, 14, 1487–1509, http://doi.org/10.1175/1520-0469(1985)042<1487:ROTAMV>2.0.CO;2, 1985

Lu, J.: Characteristics of Lightning Activity and Thermodynamic-Microphysical Mechanisms During the Organization Process of Squall Lines in Atmospheric Physics and Atmospheric Environment, Beijing: Institute of Atmospheric Physics, Chinese Academy of Sciences, 2022.

Shimada, S., Ohsawa, T., Chikaoka, S., et al.: Accuracy of the wind speed profile in the lower PBL as simulated by the WRF model, Sola, 7, 109–1112, http://doi.org/10.2151/sola.2011-028, 2011.

Pedro, A. J., Dudhia, J., Navarro, J.: On the surface wind speed probability density function over complex terrain, Geophysical Research Letters, 38, 22, L22803, http://doi.org/10.1029/2011GL049669, 2011.

Zhang, B., Liu, S., Ma, Y.: Effects of MYJ and YSU schemes on the simulation of meteorological elements in WRF boundary layer, Journal of Geophysics, 55, 7, 2239–2248, http://doi.org/ 10.6038/j.issn.0001-5733.2012.07.010, 2012.

Zheng, Y., Liu, S., Miao, Y., et al.: The influence of different terrain correction methods in the boundary layer parameterization scheme of YSU on the surface wind speed and temperature simulation, Journal of Geophysics, 59, 3, 803–815, http://doi.org/CNKI:SUN:DQWX.0.2016-03-004, 2016.

Klemp, J. B.: Dynamics of tornadic thunderstorms, Ann.rev.fluid Mech, 19, 1, 369–402, http://doi.org/10. 1146 /annurev.Fl.19. 010187. 002101, 1987.

Weisman, M. L., Davis, C. A.: Mechanisms for the generation of mesoscale vortices within quasi-linear convective systems, J .Atmos. Sci., 55, 16, 2603–2622, http://doi.org/10.1175/1520-0469(1998)0552.0.CO;2, 1998.

Gallus, W. A., Bresch, J. F.: Comparison of impacts of WRF dynamic core, physics package, and initial conditions on warm season rainfall forecasts, Monthly Weather Review, 134, 9, 2632–2641, http://doi.org/10. 1175 /mwr3198.1, 2006.

Jankov, I., Gallus, W. A., Segal. M., et al.: Influence of initial conditions on the WRF-ARW Model QPF response to physical parameterization changes, Wea Forecasting, 22, 3, 501–519, http://doi.org/10.1175/WAF998.1, 2007.

**1. The abstract does not mention what type of simulation or software/scheme was used.**

**Response:** Thank you for your feedback. We have revised the abstract to include the information that the Weather Research and Forecasting (WRF) numerical model was used in our study.

**2. The last sentence in the abstract has a good intention but ends up being a bit vague and repetitive; I'd recommend rewriting the end of the abstract to highlight the importance of the new understanding.**

**Response:** Thank you for your constructive suggestion. We have revised the last sentence of the abstract to highlight the importance of our new understanding. The revised sentence now reads: "Our findings provide crucial scientific insights for refined forecasting and early warning and risk assessment of lightning disasters, strategy formulation for urban disaster prevention and mitigation, as well as resilient city planning and development."

**3. The introduction makes it unclear if the previous studies are based on observations or simulations. This is important, as it is unclear from the texts if it's novel to do this kind of simulation.**

**Response:** Thank you for your valuable feedback. I have revised the introduction to clarify the research methods used in the previous studies. Specifically, I have added information on simulations for each study. Furthermore, I have summarized the limitations of the existing simulation studies to highlight the novelty and innovation of our research. Line 39-55 in the revised manuscript:

"Scholars have explored the dynamic effects of urban underlying surfaces through numerical simulations. The results from global and regional climate models show that the urban rough underlying surface can alter the horizontal wind field, enhancing convergence and upward movement in the upstream direction (Jin & Shepherd, 2005), which, to some extent, facilitates the development of thunderstorm systems (Yin et al.,

2020). The "climbing" upward airflow movement, as simulated by the WRF model (Zhu et al., 2016), exhibits a relatively weak intensity, insufficient to significantly alter the organizational processes of thunderstorms. Moreover, through urban boundary layer model simulations, researcher have also discovered that when thunderstorms pass over cities, the dynamic effect of the urban underlying surface can lead to the bifurcation and movement around of thunderstorm systems (Bernstein & LeRoy, 1990). This phenomenon is known as the barrier effect (Stallins & Bentley, 2006). The aforementioned simulation work has made valuable explorations into studying the dynamic effects of urban underlying surfaces. However, the current research only employs simulation scenarios with and without an urban underlying surface, without delving into the detailed characteristics of urban spatial configuration. It is worth noting that, within the spatial configuration of the urban underlying surface, city size is recognized as a crucial factor impacting thunderstorm processes (Kingfield et al., 2018). Additionally, building density also demonstrates a tendency to alter urban lightning activities (Stallins & Bentley, 2006). Therefore, it is necessary to comprehensively consider the spatial configuration characteristics of the urban underlying surface and continue to explore the influence mechanisms of the urban underlying surface on thunderstorm processes and associated lightning activities."

4. **Fig. 2: What do other colors mean? E.g., green, light green, and blue**

**Response:** Thank you for your question. In Figure 2, green represents high-density vegetation, while light green indicates low-density vegetation. Blue signifies water bodies. I have now added these explanations to the figure for clarity.

[Figure]

**Figure 2: Terrain height distribution and the building types of the WRF mesoscale numerical model.**

**5. There is a wide range of data from observations, but the simulation was performed on one specific day. How was that day chosen?**

**Response:** Thank you for your constructive suggestion.

Many years of observations in Beijing have shown that thunderstorms moving from the mountains area of Beijing are the main thunderstorm systems affecting the built-up area of Beijing (Chen et al., 2011; Wang et al., 2021). This article firstly counted the moving direction of all thunderstorms passing through Beijing. The statistical results showed that, during the eight-year period, all thunderstorms occurred in the built-up area, of which 56.6% propagated from northwest, 31.0% propagated from northeast and 12.4% propagated from southwest. Therefore, this paper took the SSW–NNE line (the green solid line in Fig. R4) as the boundary, defining the west and the east of the boundary as upwind and downwind, respectively.

[Figure]

**Figure R4: Rose diagram of propagation direction of thunderstorms in the built-up area of Beijing. The red line represented the number of thunderstorms propagating from different directions, and the green line**

**represented the boundary between upwind and downwind.**

Utilizing existing research (Bornstein & LeRoy, 1990; Dou et al., 2015), it has been noted that when thunderstorm systems traverse urban built-up areas, they exhibit a bifurcation and deflection pattern due to the barrier effect. This paper categorizes thunderstorm systems that moved into Beijing from the upwind direction between 2010 and 2017 into two types: those exhibiting significant barrier effects, designated as BT, and those classified as non-bifurcated thunderstorms, labeled as NBT. As shown in Figure S3, the year with the highest number of BT was 2013, with eight events, accounting for 23.5% of the total thunderstorms; the lowest number of BT was observed in 2010, with two events, representing 15.4% of the total thunderstorms. These results indicated that the barrier effect of urban underlying surface was a prevalent phenomenon in long-term thunderstorm observations. Furthermore, as the thunderstorm passed over the built-up area, the evolution of near-surface cold pools and convergence lines could serve as diagnostic indicators to understand how the urban underlying surface affected the thunderstorm process and CG activities. It worth noting that, among all thunderstorms occurring in the built-up area, the thunderstorm on July 13, 2017 (referred to as case "0713") produced the strongest CG activities, with flash rates of 811.6 fl/h. Therefore, this paper selects the case "0713", which exhibited the most significant blocking effect, as the study case of simulation.

[Figure]

**Figure S3: Interannual variation of bifurcated thunderstorms (BT) and non-bifurcated thunderstorms**

(NBT) in the built-up area of Beijing.

**Reference:**

Bornstein, R., and LeRoy M.: Urban barrier effect on convective and frontal thunderstorms. Preprints, Fourth Conf. on Mesoscale Processes, Boulder, CO, American Meteorological Society, 120–121, 1990.

Chen S., Wang, Y., Zhang, W., & Chen, M.: Intensifying Mechanism of the Convective Storm Moving from the Mountain to the Plain over Beijing Area. Meteorological Monthly, 3, 7, 802–813, http://doi.org/10.7519/j.issn.1000-0526.2011.7.004, 2011.

Dou J., Wang, Y., Bornstein, R., and Miao, S.: Observed Spatial Characteristics of Beijing Urban Climate Impacts on Summer Thunderstorms, Journal of Applied Meteorological Science, 54, 1, 94–105, https://doi.org/10.1175/JAMC-D-13-0355.1, 2015.

Kingfield, D. M., Calhoun, K. M., de Beurs, K. M., & Henebry, G. M.: Effects of City Size on Thunderstorm Evolution Revealed through a Multiradar Climatology of the Central United States, 57, 295–317, https://doi.org/10.1175/JAMC-D-16-0341.1, 2018.

Schmid, P. E., & Niyogi, D.: Impact of city size on precipitation-modifying potential. Geophys. Res. Lett., 40, 5263–5267. https://doi.org/10.1002/grl.50656, 2013.

Stallins, J.A.: Bentley, M.L. Urban lightning climatology and GIS: An analytical framework from the case study of Atlanta, Georgia, Appl. Geogr., 26, 242–259. https://doi.org/10.1016/j.apgeog.2006.09.008, 2006.

**6. Is there any way of visually depicting the experiments shown in Table 1, as in Figure 2?**

**Response:** Thank you for your valuable suggestion. The various sensitivity experiments in our study were indeed conducted directly within WRF through parameter settings. Visualizing these experiments, as suggested, would greatly enhance the understanding of our simulation processes for readers. We will strive to

incorporate such visualizations in our future work.

**7. Fig. 3: Why don't you keep the same limits in the color bar? Also, why are other cities presented here? Is it just to have a first idea of the effects of topography? If so, the description in the text should prevent us from talking in detail about all these regions.**

**Response:** Thank you for your valuable feedback. I apologize for any lack of clarity in our manuscript. To clarify, the focus of this paper is on the impact of urban underlying surfaces on the spatial distribution of lightning within cities. Regarding the color bar, we chose different limits for each city due to the significant variation in lightning density among them. This approach allows for a clearer observation of the internal lightning distribution characteristics within each city.

Furthermore, the inclusion of Zhangjiakou and Tianjin in comparison to Beijing serves to illustrate the potential influence of different city sizes on lightning activity. This comparison also sets the stage for the subsequent sensitivity simulation experiments related to city size.

We have revised the text to ensure that these points are more clearly communicated. Thank you again for your attention and constructive comments.

**8. L167 This paragraph just repeats the information in Fig. 4b; I don't think it's particularly useful.**

**Response:** I apologize for the unclear presentation. I have removed the redundant paragraph and provided a revised description of Figure S2 in lines 182-186.

"This section summarized the spatial distribution of CG density and various LCZs in Beijing, aiming to understand the relationship between CG activities and the urban underlying surface. The statistical results of CG density for different types of LCZ were illustrated in Figure S2. The highest average CG density is observed in LCZ1, with a value of 3.7 fl/km², while the lowest average CG density is found in LCZ6, at 2.8 fl/km²."

**9. L212 While this causal relationship is what motivates the paper, is this a conclusion that can be drawn from this analysis alone? How can you conclude that only the terrain is responsible for that evolution? Is it because of the speed of the process?**

**Response:** Thank you for your insightful comments. I apologize for the overly absolute statements in the previous paragraph, which were indeed inappropriate for a case study analysis. You are correct in pointing out that the causal relationship should not be solely based on this individual analysis.

Upon further examination, including the statistical analysis of all summer thunderstorms in Beijing (Fig. S3), we found that the barrier effect of the urban underlying surface is a prevalent phenomenon in long-term thunderstorm observations. Combining both the case study and statistical analysis, we have revised our conclusion to suggest that the urban underlying surface may be an important factor influencing thunderstorm bifurcation.

Once again, I apologize for any confusion or misinterpretation caused by our previous statements. We appreciate your guidance in improving the accuracy and rigor of our paper.

**10. Fig. 6a: What do the red and blue boxes represent?**

**Response:** I apologize for the lack of clarity in Figure 6a. The red and blue boxes in Figure 6a represent the wind direction variation with height at different times. The red box indicates clockwise variation at 08:00 BJT, suggesting weak warm advection, while the blue box shows counterclockwise variation at 14:00 BJT, indicating strong cold advection.

**11. L241 Some of us are unfamiliar with the Taihang Mountain, is it highlighted in the Figure?**

**Response:** Thank you for your excellent suggestion. We have added the highlight of the Taihang Mountain in Figure 1b as per your advice.

[Figure]

Figure 1: Overview of the study area (a). Topography and built-up area in Beijing megacity (b).

**12. L259 This is an interesting classification. Did the BT events show any particular atmospheric feature?**

**Response:** Thank you very much for your excellent suggestion. We have conducted a preliminary exploration of the relationship between synoptic backgrounds and BT (bifurcation of thunderstorm) events here. While our analysis does not directly address specific atmospheric features associated with BT events, it lays a foundation for future research in this direction.

The synoptic backgrounds of thunderstorms in Beijing were classified into the weak synoptic background and strong synoptic background based on previous studies (Dixon & Mote, 2003; Stallins & Bentley, 2006). The features of two synoptic backgrounds are shown as followings.

[Figure]

Figure R3: Synoptic patterns of 500 hPa (a) and 850 hPa (b) at 08:00 on June 8, 2014, (black solid lines are

Fig. R5a shows that the northeast cold vortex was very active at 08:00 on June 8, 2014. At 500 hPa, the Beijing area was behind trough, controlled by dry and cold northwestern flow, with specific humidity of about 2 g/kg. At 850 hPa (Fig R5b), there was weak wind shear around Beijing, with specific humidity below 6 g/kg. The lower layers within a warm dry tongue. The sounding curve at 08:00 hours shows that the low-level air was dry and warm, which was not conducive to the development of thunderstorm, and the convective available potential energy (CAPE) was 109.2 J/kg. With the accumulation of unstable energy, the CAPE value increased to 218.3 J/kg at the start of the thunderstorm (14:00). Therefore, if there was no synoptic-scale backward-tilting trough affecting the study area and no significant wind shear at lower levels, the synoptic background for this thunderstorm was referred to as a weak synoptic background in this paper.

[Figure]

Figure R6: Synoptic patterns of 500 hPa (a) and 850 hPa (b) at 08:00 on July 13, 2017 (black solid lines are contours; red dashed lines are isotherms, shaded areas are specific humidity fields).

Fig. R6a presents that the Beijing area was in front of trough at 08:00 on July 13, 2017, with a specific humidity of about 3.5 g/kg. At 850 hPa (Fig. R6b), an area of high-pressure existed south of 45°N. To the northwest of the high pressure zone, there was a continuous transport of warm and humid airflows, with significant wind shear

around 41°N. The specific humidity in the Beijing area exceeded 12 g/kg, forming an unstable stratification characterized by dryness above and wetness below. The sounding curve at 08:00 shows that the CAPE value was 3783.5 J/kg, and the atmospheric stratification was unstable. Before the onset of the thunderstorm, the ground specific humidity was about 13 g/kg, and CAPE value reached 4333.1 J/kg, providing favorable conditions for strong convection. Therefore, if the study area was influenced by a synoptic-scale backward-tilting trough, with abundant warm and humid airflow cooperating and significant wind shear at lower level, the synoptic background for this thunderstorm was referred to as a strong synoptic background in this paper.

Fig. R7a shows the spatial patterns of lightning with the weak synoptic background. There were several lightning-intensive areas in the built-up areas, and the CG flashes density in the city center reached 3.9 fl/km$^2$, so the weak synoptic background might be unfavorable for BT thunderstorm events. Under the condition of strongly unstable stratification, bifurcations and moving around will occur when frontal thunderstorms pass through cities (Bornstein & LeRoy, 1990). The down-to-hill thunderstorms with the strong synoptic system formed lightning-intensive areas in the upwind and downwind of built-up areas (Fig. R7b), with the significant lightning-sparse areas (S) in the city center. Local unstable air masses with low energy and synoptic-scale frontal systems have different thermodynamic processes and influence the spatial patterns of CG activity (Stallins, 2006). In summary, the strong synoptic background might be an important condition for the BT thunderstorm events.

[Figure]

**Figure R7: The spatial patterns of long series of summertime CG flashes density for (a) weak synoptic background, and (b) strong synoptic background.**

Reference:

Stallins, J.A.: Bentley, M.L. Urban lightning climatology and GIS: An analytical framework from the case study of Atlanta, Georgia. Appl. Geogr, 26, 242–259. https://doi.org/10.1016/j.apgeog.2006.09.008, 2006.

Dixon, P. G., Mote, T. L.: Patterns and causes of Atlanta's urban heat island—initiated precipitation, J. Appl. Meteor., 42, 9, 1273–1284, doi: https://doi.org/10.1175/1520-0450(2003)0422.0.CO;2, 2003.

**13. L274 Since this paper has a numerical simulation, the analysis should be deeper. If there are errors, we should have some sensitivity analysis to better understand how the model works.**

**Response:** Thank you once again for your comments on the simulation errors. Please refer to the response to the second question in major comment 2 for a detailed reply regarding this issue.

**14. Fig. 8: What are the units for vertical velocity? What are the thick red lines in the x-axis of a-b? The red area in 8a(left) is not easy to distinguish. It is also**

**not easy to see clearly the isolines of vertical velocity. How about shading the areas lower than -1 K to see the cold pool clearly?**

**Response:** I sincerely apologize for the oversights and ambiguities in our previous submission. Regarding your comments, the units for vertical velocity have been corrected to m/s and have been supplemented in the text. Additionally, the isolines of vertical velocity have been bolded for better clarity. The thick red lines on the x-axis of Figures 8a and 8b represent the urban underlying surface, and an example explanation has been included in the text for clarification. To enhance distinguishability, the red boundary in Figure 8a (left) has been changed to black. Your suggestion to shade the areas lower than -1 K to clearly visualize the cold pool is highly appreciated. To further highlight the continuity of the cold pool core and its surrounding density currents, we have outlined the cold pool core with dashed purple lines in Figure 8a (right) and provided an explanation in the legend. Thank you for your valuable feedback.

[Figure]

**Figure 8: Influence of the scale of the built-up area on near-surface cold pool outflow simulated by WRF. Figures (a) to (c) represent experiments 1 to 3, respectively. In each set of subfigures, the left subfigure represents the horizontal thermal-dynamical field, and the middle and right subfigures depict the cross-sections along line CD and EF respectively, including perturbation potential temperature (shaded, unit: K), east-west circulation (vector), and vertical velocity (isoline, unit: m/s).**

**15.  Fig. 9: Red arrows are hard to distinguish. Again, the red line over the x-axis is not explained.**

**Response:** I apologize for the difficulties you encountered in distinguishing the red arrows and for the lack of explanation regarding the red line over the x-axis in Figure 9. To address your concerns, the red arrows have been made bolder for better visibility. Additionally, I have clarified in the figure caption that the red line over the x-axis

represents the urban underlying surface. Thank you for your valuable feedback.

[Figure]

Figure 9: Influence of urban underlying surface on near-surface convergence zones simulated by WRF. Figures (a) to (e) represent experiments 1 to 5, respectively. In each set of subfigures, the left subfigure represented the horizontal wind field and divergence field, while the right subfigure depicted the cross-section of vertical velocity along line GH.

**16. L340 It is unclear from the text how much of this mechanism has already been studied and explained. What part of this is novel in the study?**

**Response:** Thank you for your insightful comments. Regarding the current understanding of how urban underlying surfaces influence lightning activity, a systematic conceptual model has not yet been developed. Previous studies have primarily focused on the impact of urban areas on the spatial distribution of precipitation (Dou et al., 2015; Qian et al., 2022; Yang et al., 2024). For example, Dou et al. (2024) have proposed hypotheses regarding the causes of urban rainfall anomalies: (i) Under weak UHI conditions, it is the urban dynamical effect that played the dominant role. The surface flow was blocked by the urban rough surface,

inducing updrafts around the central urban area (CUA) and downdraft over the CUA. As the thunderstorm passed, its convergence line broke over the CUA. The precipitation thus bifurcated; (ii) Under strong UHI conditions, it is the urban thermal factor that played the dominant role, overshadowing the urban dynamical impacts which still existed. The intensified UHI led to a more organized UHI circulation, with stronger updrafts over the CUA. When the storm arrived, the pre-existing updraft and convergence was conducive to convection initiation and enhancement. The precipitation thus concentrated over CUA and strengthened. Employment of a cooling tower scheme improved the simulated precipitation. This study thus represents an initial attempt to differentiate urban thermal and dynamic impacts on thunderstorms.

However, the timing of precipitation peaks often lags behind that of lightning, resulting in a mismatch between the locations of peak precipitation and lightning (Wu et al., 2017, 2018; Li et al., 2017). Therefore, these hypotheses can only serve as a reference for understanding the mechanisms underlying the spatial characteristics of urban lightning. More importantly, our study shifts the focus from the UHI to the spatial configuration characteristics of the urban underlying surface (scale and density), which represents one of the novel aspects of our research.

Reference:

Dou, J., Wang, Y., Bornstein, R., and Miao, S.: Observed Spatial Characteristics of Beijing Urban Climate Impacts on Summer Thunderstorms, Journal of Applied Meteorological Science, 54, 1, 94–105, https://doi.org/10.1175/JAMC-D-13-0355.1, 2015.

Dou, J., Bornstein, R., Sun, J., and Miao, S.: Impacts of urban heat island intensities on a bifurcating thunderstorm over Beijing, Urban climate, 55, 101955, https://doi.org/10.1016/j.uclim.2024.101955, 2024.

Qian, Y., Chakraborty, T.C., Li, J., Li, D., He, C., Sarangi, C., Chen, F., Yang, X., Leung, R.: Urbanization impact on regional climate and extreme weather: Current understanding, uncertainties, and future research directions. Adv. Atmos. Sci., 39, 6, 819−860, https://doi.org/10.1007/s00376-021-1371-9, 2022.

Yang, L., Yang, Y., Shen, Y., Yang, J., Zheng, G., Smith, L., & Niyogi, D.: Urban development pattern's influence on extreme rainfall occurrences, Nature Communications, 15, 1, https://doi.org/10.1038/s41467-024-48533-5, 2024.

Wu, F., Cui, X., Zhang, D.: A lightning-based nowcast-warning approach for short-duration rainfall events: Development and testing over Beijing during the warm seasons of 2006-2007, Atmos. Res., 205, 2-17, https://doi.org/10.1016/j.atmosres.2018.02.003, 2018.

Li, H., Cui, X., Zhang, D.: A statistical analysis of hourly heavy rainfall events over the Beijing metropolitan region during the warm seasons of 2007-2014, Int. J. Climatol., 37, 4027-4042, https://doi.org/10.1002/joc.4983, 2017.

Wu, F., Cui, X., Zhang, D., Qiao, L.: The relationship of lightning activity and short-duration rainfall events during warm seasons over the Beijing metropolitan region, Atmos. Res., 195, 31-43, https://doi.org/10.1016/j.atmosres.2017.04.032, 2017.

---

## Author Response (AR4)

**Response to Reviewer Comments**

Dear Reviewer and Editors:

We are sincerely grateful to the editor and reviewer for their valuable time for reviewing our manuscript. The comments are very helpful and valuable, and we have addressed the issues raised by the reviewer in the revised manuscript. Please find our point-by-point response (in blue text) to the comments (in black text) raised by the reviewer. We have revised the paper according to your comments (highlighted in red text of the revised manuscript).

**The authors have attended many of my previous comments in their response, improving the clarity of the manuscript. However, some key responses were not added to the final version. Therefore, I suggest them to include these details.**

**Response:** Thank you for your recognition of our work and for your valuable feedback. As per your request, we have undertaken revisions throughout the manuscript.

**Minor suggestions:**

**1. You mentioned that the surface resolution data is 10 m, but the rest of the model is 200 m. How do they interact? (That was my previous question; sorry if it wasn't clear enough)**

**Response:** Thank you for your question. To clarify, the 10 m underlying surface data were used to accurately describe urban morphological parameters. These data were matched the nested grid before being implemented in the UCM. We have added clarification in the revised manuscript (lines 141-142). Please let us know if further explanation would be helpful.

**2. It'd be useful to include part of the response to the question regarding**

**lightning models in WRF in your manuscript.**

**Response:** Thank you for the suggestion. We have added the lightning model details to the manuscript (lines 131-133).

**3. Please include the explanation of the chosen day in the manuscript, as in the response**

**Response:** Thank you for the suggestion. We have added the chosen day details to the manuscript (lines 147-150).

**4. The response to including a figure for Table 1 was not attended. Is it because it is too much work to do it?**

**Response:** Thank you for your continued interest in improving our manuscript's clarity. We acknowledge that creating a visual representation of Table 1 would indeed be valuable, and we regret not including it initially. The omission was not due to workload considerations, but rather because our parameter modifications in WRF-UCM were primarily numerical adjustments that don't lend themselves to intuitive graphical representation. Future work will explore effective methods to visually represent such numerical parameter experiments.

**5. Please include part of the response regarding the last comment on the mechanism explanation.**

**Response:** Thank you for the suggestion. We have added the mechanism explanation details to the manuscript (lines 338-342).

**6. Text typos. L45 "researchers"**

**Response:** Thank you for catching this. We have corrected and conducted a full proofreading of the manuscript to ensure no other typos remain.